# Suppressing quantum errors by scaling a surface code logical qubit

Google Quantum AI*

Practical quantum computing will require error rates well below those achievable with physical qubits. Quantum error correction[1,2] offers a path to algorithmically relevant error rates by encoding logical qubits within many physical qubits, for which increasing the number of physical qubits enhances protection against physical errors. However, introducing more qubits also increases the number of error sources, so the density of errors must be sufficiently low for logical performance to improve with increasing code size. Here we report the measurement of logical qubit performance scaling across several code sizes, and demonstrate that our system of superconducting qubits has sufficient performance to overcome the additional errors from increasing qubit number. We find that our distance-5 surface code logical qubit modestly outperforms an ensemble of distance-3 logical qubits on average, in terms of both logical error probability over 25 cycles and logical error per cycle ((2.914 ± 0.016)% compared to (3.028 ± 0.023)%). To investigate damaging, low-probability error sources, we run a distance-25 repetition code and observe a $1.7 \times 10^{-6}$ logical error per cycle floor set by a single high-energy event ($1.6 \times 10^{-7}$ excluding this event). We accurately model our experiment, extracting error budgets that highlight the biggest challenges for future systems. These results mark an experimental demonstration in which quantum error correction begins to improve performance with increasing qubit number, illuminating the path to reaching the logical error rates required for computation.

Since Feynman's proposal to compute using quantum mechanics[3], many potential applications have emerged, including factoring[4], optimization[5], machine learning[6], quantum simulation[7] and quantum chemistry[8]. These applications often require billions of quantum operations[9–11] and state-of-the-art quantum processors typically have error rates around $10^{-3}$ per gate[12–17], far too high to execute such large circuits. Fortunately, quantum error correction can exponentially suppress the operational error rates in a quantum processor, at the expense of temporal and qubit overhead[18,19].

Several works have reported quantum error correction on codes able to correct a single error, including the distance-3 Bacon–Shor[20], colour[21], five-qubit[22], heavy-hexagon[23] and surface[24,25] codes, as well as continuous variable codes[26–29]. However, a crucial question remains of whether scaling up the error-correcting code size will reduce logical error rates in a real device. In theory, logical errors should be reduced if physical errors are sufficiently sparse in the quantum processor. In practice, demonstrating reduced logical error requires scaling up a device to support a code that can correct at least two errors, without sacrificing state-of-the-art performance. In this work we report a 72-qubit superconducting device supporting a 49-qubit distance-5 ($d = 5$) surface code that narrowly outperforms its average subset 17-qubit distance-3 surface code, demonstrating a critical step towards scalable quantum error correction.

## Surface codes with superconducting qubits

Surface codes[30–34] are a family of quantum error-correcting codes that encode a logical qubit into the joint entangled state of a $d \times d$ square of physical qubits, referred to as data qubits. The logical qubit states are defined by a pair of anti-commuting logical observables $X_L$ and $Z_L$. For the example shown in Fig. 1a, a $Z_L$ observable is encoded in the joint $Z$-basis parity of a line of qubits that traverses the lattice from top to bottom, and likewise an $X_L$ observable is encoded in the joint $X$-basis parity traversing left to right. This non-local encoding of information protects the logical qubit from local physical errors, provided we can detect and correct them.

To detect errors, we periodically measure $X$ and $Z$ parities of adjacent clusters of data qubits with the aid of $d^2 - 1$ measure qubits interspersed throughout the lattice. As shown in Fig. 1b, each measure qubit interacts with its neighbouring data qubits to map the joint data qubit parity onto the measure qubit state, which is then measured. Each parity measurement, or stabilizer, commutes with the logical observables of the encoded qubit as well as every other stabilizer. Consequently, we can detect errors when parity measurements change unexpectedly, without disturbing the logical qubit state.

A decoder uses the history of stabilizer measurement outcomes to infer likely configurations of physical errors on the device. We can then

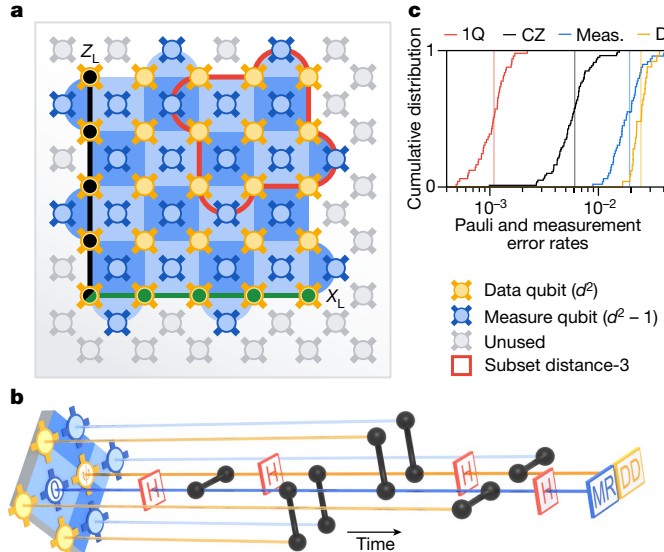

**Fig. 1 | Implementing surface code logical qubits. a**, Schematic of a 72-qubit Sycamore device with a distance-5 surface code embedded, consisting of 25 data qubits (gold) and 24 measure qubits (blue). Each measure qubit is associated with a stabilizer (blue coloured tile, dark: $X$, light: $Z$). Representative logical operators $Z_L$ (black) and $X_L$ (green) traverse the array, intersecting at the lower-left data qubit. The upper right quadrant (red outline) is one of four subset distance-3 codes (the four quadrants) that we compare to distance-5. **b**, Illustration of a stabilizer measurement, focusing on one data qubit (labelled ψ) and one measure qubit (labelled 0), in perspective view with time progressing to the right. Each qubit participates in four CZ gates (black) with its four nearest neighbours, interspersed with Hadamard gates (H), and finally, the measure qubit is measured and reset to $|0\rangle$ (MR). Data qubits perform dynamical decoupling (DD) while waiting for the measurement and reset. All stabilizers are measured in this manner concurrently. Cycle duration is 921 ns, including 25-ns single-qubit gates, 34-ns two-qubit gates, 500-ns measurement and 160-ns reset (see Supplementary Information for compilation details). The readout and reset take up most of the cycle time, so the concurrent data qubit idling is a dominant source of error. **c**, Cumulative distributions of errors for single-qubit gates (1Q), CZ gates, measurement (Meas.) and data qubit dynamical decoupling (idle during measurement and reset), which we refer to as component errors. The circuits were benchmarked in simultaneous operation using random circuit techniques, on the 49 qubits used in distance-5 and the 4 CZ layers from the stabilizer circuit[38,59] (see Supplementary Information). Vertical lines are means.

determine the overall effect of these inferred errors on the logical qubit, thus preserving the logical state. Most surface code logical gates can be implemented by maintaining logical memory and executing different sequences of measurements on the code boundary[35–37]. Thus, we focus on preserving logical memory, the core technical challenge in operating the surface code.

We implement the surface code on an expanded Sycamore device[38] with 72 transmon qubits[39] and 121 tunable couplers[40,41]. Each qubit is coupled to four nearest neighbours except on the boundaries, with mean qubit coherence times $T_1 = 20$ μs and $T_{2,CPMG} = 30$ μs, in which CPMG represents Carr–Purcell–Meiboom–Gill. As in ref. [42], we implement single-qubit rotations, controlled-$Z$ (CZ) gates, reset and measurement, demonstrating similar or improved simultaneous performance as shown in Fig. 1c.

The distance-5 surface code logical qubit is encoded on a 49-qubit subset of the device, with 25 data qubits and 24 measure qubits. Each measure qubit corresponds to one stabilizer, classified by its basis ($X$ or $Z$) and the number of data qubits involved (weight, 2 or 4). Ideally, to assess how logical performance scales with code size, we would compare distance-5 and distance-3 logical qubits under identical noise.

Although device inhomogeneity makes this comparison difficult, we can compare the distance-5 logical qubit to the average of four distance-3 logical qubit subgrids, each containing nine data qubits and eight measure qubits. These distance-3 logical qubits cover the four quadrants of the distance-5 code with minimal qubit overlap, capturing the average performance of the full distance-5 grid.

In a single instance of the experiment, we initialize the logical qubit state, run several cycles of error correction, and then measure the final logical state. We show an example in Fig. 2a. To prepare a $Z_L$ eigenstate, we first prepare each data qubit in $|0\rangle$ or $|1\rangle$, an eigenstate of the $Z$ stabilizers. The first cycle of stabilizer measurements then projects the data qubits into an entangled state that is also an eigenstate of the $X$ stabilizers. Each cycle contains CZ and Hadamard gates sequenced to extract $X$ and $Z$ stabilizers simultaneously, and ends with the measurement and reset of the measure qubits. In the final cycle, we also measure the data qubits in the $Z$ basis, yielding both parity information and a measurement of the logical state. Preparing and measuring $X_L$ eigenstates proceeds analogously. The instance succeeds if the corrected logical measurement agrees with the known initial state; otherwise, a logical error has occurred.

Our stabilizer circuits contain a few modifications to the standard gate sequence described above (see Supplementary Information), including phase corrections to correct for unintended qubit frequency shifts and dynamical decoupling gates during qubit idles[43]. We also remove certain Hadamard gates to implement the $ZXXZ$ variant of the surface code[44,45], which helps symmetrize the $X$- and $Z$-basis logical error rates. Finally, during initialization, the data qubits are prepared into randomly selected bitstrings. This ensures that we do not preferentially measure even parities in the first few cycles of the code, which could artificially lower logical error rates owing to bias in measurement error (see Supplementary Information).

## Error detectors

After initialization, parity measurements should produce the same value in each cycle, up to known flips applied by the circuit. If we compare a parity measurement to the corresponding measurement in the preceding cycle and their values are inconsistent, a detection event has occurred, indicating an error. We refer to these comparisons as detectors.

The detection event probabilities for each detector indicate the distribution of physical errors in space and time while running the surface code. In Fig. 2, we show the detection event probabilities in the distance-5 code (Fig. 2b,c) and the distance-3 codes (Fig. 2d,e) running for 25 cycles, as measured over 50,000 experimental instances. For the weight-4 stabilizers, the average detection probability is $0.185 \pm 0.018$ ($1\sigma$) in the distance-5 code and $0.175 \pm 0.017$ averaged over the distance-3 codes. The weight-2 stabilizers interact with fewer qubits and hence detect fewer errors. Correspondingly, they yield a lower average detection probability of $0.119 \pm 0.012$ in the distance-5 code and $0.115 \pm 0.008$ averaged over the distance-3 codes. The relative consistency between code distances suggests that growing the lattice does not substantially increase the component error rates during error correction.

The average detection probabilities exhibit a relative rise of 12% for distance-5 and 8% for distance-3 over 25 cycles, with a typical characteristic risetime of roughly 5 cycles (see Supplementary Information). We attribute this rise to data qubits leaking into non-computational excited states and anticipate that the inclusion of leakage-removal techniques on data qubits would help to mitigate this rise[42,46–48]. We reason that the greater increase in detection probability in the distance-5 code is due to increased stray interactions or leakage from simultaneously operating more gates and measurements.

We test our understanding of the physical noise in our system by comparing the experimental data to a simulation. We begin with a

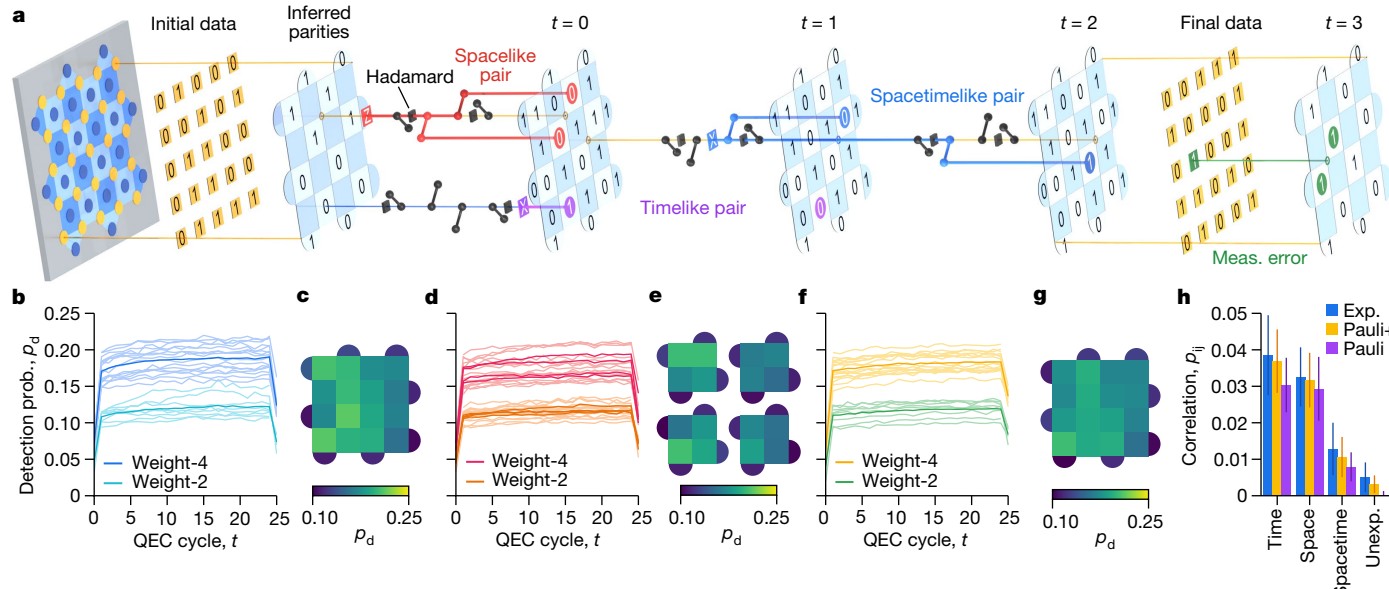

**Fig. 2 | Error detection in the surface code. a**, Illustration of a surface code experiment, in perspective view with time progressing to the right. We begin with an initial data qubit state that has known parities in one stabilizer basis (here, $Z$). We show example errors that manifest in detection pairs: a $Z$ error (red) on a data qubit (spacelike pair), a measurement error (purple) on a measure qubit (timelike pair), an $X$ error (blue) during the CZ gates (spacetimelike pair) and a measurement error (green) on a data qubit (detected in the final inferred $Z$ parities). **b**, Detection probability for each stabilizer over a 25-cycle distance-5 experiment (50,000 repetitions). Darker lines: average over all stabilizers with the same weight. There are fewer detections at timestep $t = 0$ because there is no preceding syndrome extraction, and at $t = 25$ because the final parities are calculated from data qubit measurements directly. QEC, quantum error

correction. **c**, Detection probability heatmap, averaging over $t = 1$ to 24. **d**,**e**, Similar to **b**,**c** for four separate distance-3 experiments covering the four quadrants of the distance-5 code. **f**,**g**, Similar to **b**,**c** using a simulation with Pauli errors plus leakage, crosstalk and stray interactions (Pauli+). **h**, Bar chart summarizing the detection correlation matrix $p_{ij}$, comparing the distance-5 experiment from **b** to the simulation in **f** (Pauli+) and a simpler simulation with only Pauli errors. We aggregate four groups of correlations: timelike pairs; spacelike pairs; spacetimelike pairs expected for Pauli noise; and spacetimelike pairs unexpected for Pauli noise (Unexp.), including correlations over two timesteps. Each bar shows a mean and standard deviation of correlations from a 25-cycle, 50,000-repetition dataset.

depolarizing noise simulation based on the component error information in Fig. 1c, and then extend to a Pauli simulation with qubit-specific $T_1$ and $T_{2,\text{CPMG}}$, transitions to leaked states, and stray interactions between qubits during CZ gates (see Supplementary Information). We refer to this simulation as Pauli+. Figure 2f shows that this second simulator accurately predicts the average detection probabilities, finding $0.180 \pm 0.013$ for the weight-4 stabilizers and $0.116 \pm 0.011$ for the weight-2 stabilizers, with average detection probabilities increasing 7% over 25 cycles (distance-5).

## Understanding errors through correlations

We next examine pairwise correlations between detection events, which give us fine-grained information about which types of error are occurring during error correction. Figure 2a illustrates a few examples of pairwise detections that are generated by $X$ or $Z$ errors in the surface code. Measurement and reset errors are detected by the same stabilizer in two consecutive cycles, which we classify as a timelike pair. Data qubits may experience an $X$ ($Z$) error while idling during measurement that is detected by its neighbouring $Z$ ($X$) stabilizers in the same cycle, forming a spacelike pair. Errors during CZ gates may cause a variety of pairwise detections to occur, including spacetimelike pairs that are separated in both space and time. More complex clusters of detection events arise when a $Y$ error occurs, which generates detection events for both $X$ and $Z$ errors.

To estimate the probability for each detection event pair from our data, we compute an appropriately normalized correlation $p_{ij}$ between detection events occurring on any two detectors $i$ and $j$ (refs. [42,49]; see Supplementary Information). In Fig. 2h, we show the estimated probabilities for experimental and simulated distance-5 data, aggregated

and averaged according to the different classes of pairs. In addition to the expected pairs, we also quantify how often detection pairs occur that are unexpected in a local depolarizing circuit model. Overall, the Pauli simulation systematically underpredicts these probabilities compared to experimental data, whereas the Pauli+ simulation is closer and predicts the presence of unexpected pairs, which we surmise are related to leakage and stray interactions. These errors can be especially harmful to the surface code because they can generate multiple detection events distantly separated in space or time, which a decoder might wrongly interpret as multiple independent component errors. We expect that mitigating leakage and stray interactions will become increasingly important as error rates decrease.

## Decoding and logical error probabilities

We next examine the logical performance of our surface code qubits. To infer the error-corrected logical measurement, the decoder requires a probability model for physical error events. This information may be expressed as an error hypergraph: detectors are vertices, physical error mechanisms are hyperedges connecting the detectors they trigger, and each hyperedge is assigned its corresponding error mechanism probability. We use a generalization of $p_{ij}$ to determine these probabilities[42,50].

Given the error hypergraph, we implement two different decoders: belief-matching, an efficient combination of belief propagation and minimum-weight perfect matching[51]; and tensor network decoding, a slow but accurate approximate maximum-likelihood decoder. The belief-matching decoder first runs belief propagation on the error hypergraph to update hyperedge error probabilities based on nearby detection events[51,52]. The updated error hypergraph is then decomposed

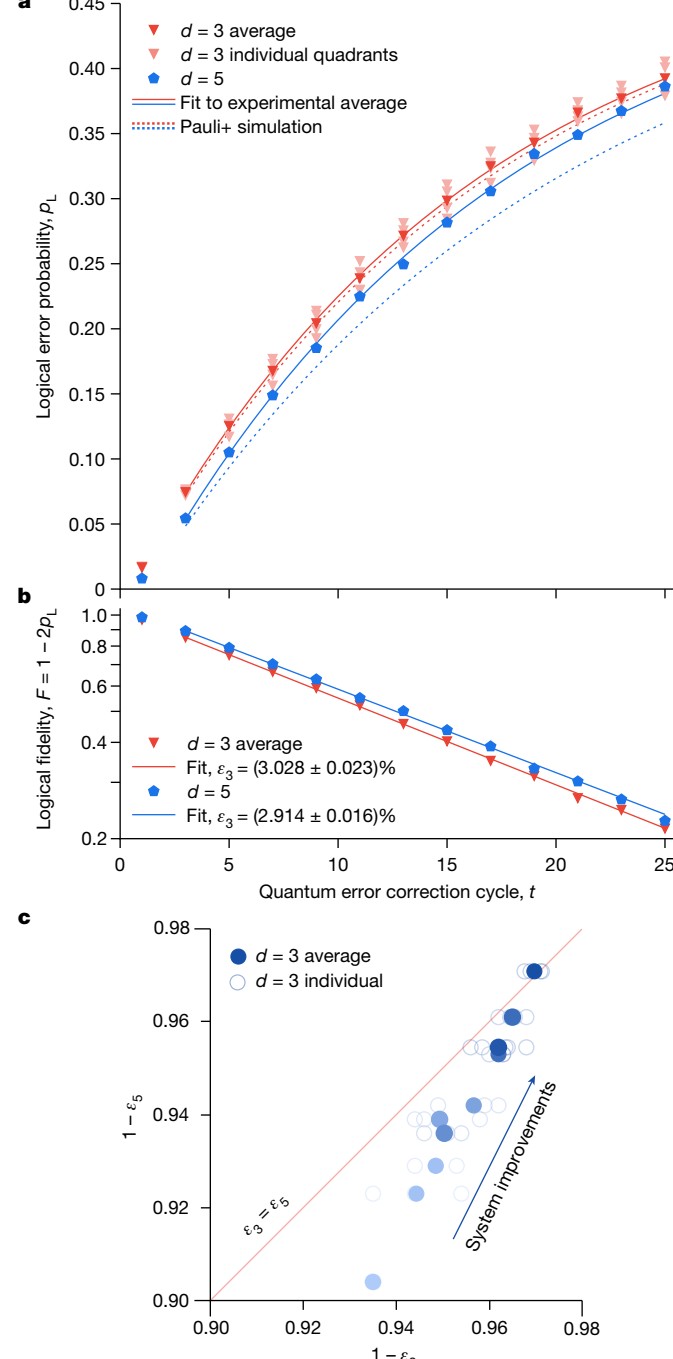

**Fig. 3 | Logical error reduction. a**, Logical error probability $p_L$ versus cycle comparing distance-5 (blue) to distance-3 (pink: four separate quadrants, red: average), all averaged over $Z_L$ and $X_L$. Each individual data point represents 100,000 repetitions. Solid line: fit to experimental average, $t = 3$ to 25 (see main text). Dotted line: comparison to Pauli+ simulation. **b**, Logical fidelity $F = 1 - 2p_L$ versus cycle, semilog plot. The datapoints and fits are the experimental averages and fits from **a**. **c**, Summary of experimental progression comparing logical error per cycle $\varepsilon_d$ (specifically plotting $1 - \varepsilon_d$) between distance-3 and distance-5, for which system improvements lead to faster improvement for distance-5 (see main text). Each open circle is a comparison to a specific distance-3 code, and filled circles average over several distance-3 codes measured in the same session. Markers are coloured chronologically from light to dark. Typical $1\sigma$ statistical and fit uncertainty is 0.02%, smaller than the points.

into a pair of disjoint error graphs, one each for $X$ and $Z$ errors[31]. These graphs are decoded efficiently using minimum-weight perfect matching[53] to select a single probable set of errors.

By contrast, a maximum-likelihood decoder considers all possible sets of errors consistent with the detection events, splits them into two groups on the basis of whether they flip the logical measurement, and chooses the group with the greater total likelihood. The two likelihoods are each expressed as a tensor network contraction[51,54,55] that exhaustively sums the probabilities of all sets of errors within each group. We can contract the network approximately, and verify that the approximation converges. This yields a decoder that is nearly optimal given the hypergraph error priors, but is considerably slower. Further improvements could come from a more accurate prior, or by incorporating more fine-grained measurement information[47,56].

Figure 3 shows a comparison of the logical error performance of the distance-3 and distance-5 codes using the approximate maximum-likelihood decoder. As the $ZXXZ$ variant of the surface code symmetrizes the $X$ and $Z$ bases, differences between the two bases' logical error per cycle are small and attributable to spatial variations in physical error rates. Thus, for visual clarity, we report logical error probabilities averaged between the $X$ and $Z$ basis; the full dataset may be found in the Supplementary Information. Note that we do not post-select on leakage or high-energy events to capture the effects of realistic non-idealities on logical performance. Over all 25 cycles of error correction, the distance-5 code realizes lower logical error probabilities $p_L$ than the average of the subset distance-3 codes.

We fit the logical fidelity $F = 1 - 2p_L$ to an exponential decay. We start the fit at $t = 3$ to avoid two phenomena that advantage the larger code: the lower detection probability during the first cycle relative to subsequent cycles (Fig. 2b,d), and the higher effective threshold caused by the confinement of errors to thin time slices in few-cycle experiments[31]. We obtain a logical error per cycle $\varepsilon_5 = (2.914 \pm 0.016)\%$ ($1\sigma$ statistical and fit uncertainty) for the distance-5 code, compared to an average of $\varepsilon_3 = (3.028 \pm 0.023)\%$ for the subset distance-3 codes, a relative error reduction of about 4%. When decoding with the faster belief-matching decoder, we fit a logical error per cycle of $(3.056 \pm 0.015)\%$ for the distance-5 code, compared to an average of $(3.118 \pm 0.025)\%$ for the distance-3 codes, a relative error reduction of about 2%. We note that the distance-5 logical error per cycle is slightly higher than those of two of the distance-3 codes individually, and that leakage accumulation may cause distance-5 performance to degrade faster than that of distance-3 as logical error probability approaches 50%.

In principle, the logical performance of a distance-5 code should improve faster than that of a distance-3 code as physical error rates decrease[33]. Over time, we improved our physical error rates, for example by optimizing single- and two-qubit gates, measurement and data qubit idling (see Supplementary Information). In Fig. 3c, we show the corresponding performance progression of distance-5 and distance-3 codes. The larger code improved about twice as fast until finally overtaking the smaller code, validating the benefit of increased-distance protection in practice.

To understand the contributions of individual components to our logical error performance, we follow ref. [42] and simulate the distance-5 and distance-3 codes while varying the physical error rates of the various circuit components. As the logical-error-suppression factor

$$\Lambda_{d/(d+2)} = \varepsilon_d / \varepsilon_{d+2} \tag{1}$$

is approximately inversely proportional to the physical error rate, we can budget how much each physical error mechanism contributes to $1/\Lambda_{3/5}$ (as shown in Fig. 4a) to assess scaling. This error budget shows that CZ error and data qubit decoherence during measurement and reset are dominant contributors.

## Algorithmically relevant error rates

Even as known error sources are suppressed in future devices, new dominant error mechanisms may arise as lower logical error rates are realized. To test the behaviour of codes with substantially lower error

# Article

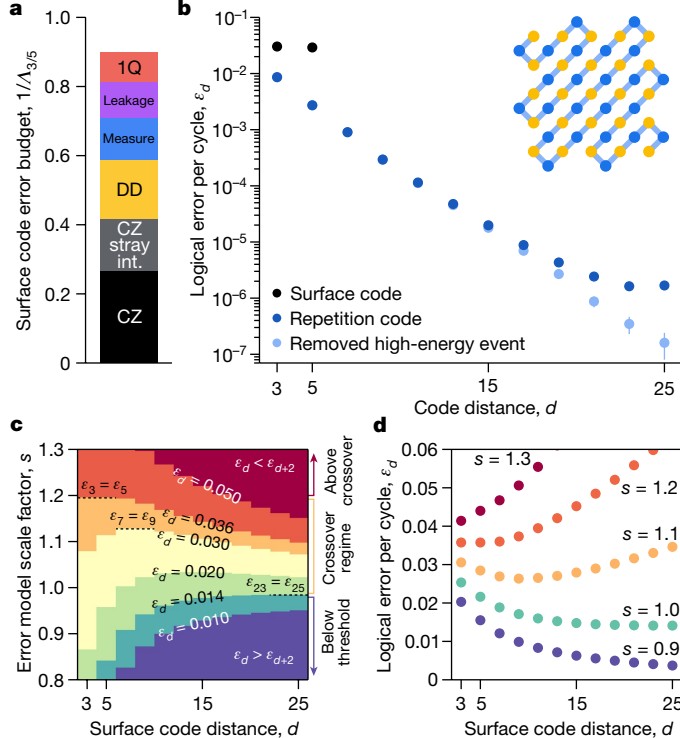

**a** Surface code error budget, $1/\Lambda_{3/5}$

Legend: 1Q, Leakage, Measure, DD, CZ stray int., CZ

**b** Logical error per cycle, $\varepsilon_d$

- Surface code
- Repetition code
- Removed high-energy event

Code distance, $d$

**c** Error model scale factor, $s$

$\varepsilon_3 = \varepsilon_5$    $\varepsilon_d < \varepsilon_{d+2}$    $\varepsilon_d = 0.050$
$\varepsilon_7 = \varepsilon_9$    $\varepsilon_d = 0.036$    $\varepsilon_d = 0.030$
$\varepsilon_d = 0.020$
$\varepsilon_d = 0.014$    $\varepsilon_{23} = \varepsilon_{25}$
$\varepsilon_d = 0.010$
$\varepsilon_d > \varepsilon_{d+2}$

Above crossover / Crossover regime / Below threshold

Surface code distance, $d$

**d** Logical error per cycle, $\varepsilon_d$

$s = 1.3$, $s = 1.2$, $s = 1.1$, $s = 1.0$, $s = 0.9$

Surface code distance, $d$

**Fig. 4 | Towards algorithmically relevant error rates. a**, Estimated error budget for the surface code, based on component errors (see Fig. 1c) and Pauli+ simulations. $\Lambda_{3/5} = \varepsilon_3/\varepsilon_5$. CZ, contributions from CZ error (excluding leakage and stray interactions). CZ stray int., CZ error from unwanted interactions. DD, dynamical decoupling (data qubit idle error during measurement and reset). Measure, measurement and reset error. Leakage, leakage during CZs and due to heating. 1Q, single-qubit gate error. **b**, Logical error for repetition codes. Inset: schematic of the distance-25 repetition code, using the same data and measure qubits as the distance-5 surface code. Smaller codes are subsampled from the same distance-25 data[42]. A high-energy event resulted in an apparent error floor around $10^{-6}$. After removing the instances nearby (light blue), error decreases more rapidly with code distance. The dataset has 50 cycles, $5 \times 10^5$ repetitions. We also plot the surface code error per cycle from Fig. 3b in black. **c**, Contour plot of simulated surface code logical error per cycle $\varepsilon_d$ as a function of code distance $d$ and a scale factor $s$ on the error model in Fig. 1c (Pauli simulation, $s = 1.0$ corresponds to the current device error model). **d**, Horizontal slices from **c**, each for a value of error-model scale factor $s$. $s = 1.3$ is above threshold (larger codes are worse), and $s = 1.2$ to 1.0 represent the crossover regime, for which progressively larger codes get better until a turnaround. $s = 0.9$ is below threshold (larger codes are better).

rates, we use the bit-flip repetition code, a one-dimensional version of the surface code. The bit-flip repetition code does not correct for phase-flip errors and is thus unsuitable for quantum algorithms. However, correcting only bit-flip errors allows it to achieve much lower logical error probabilities.

Without post-selection, we achieve a logical error per cycle of $(1.7 \pm 0.3) \times 10^{-6}$ using a distance-25 repetition code decoded with minimum-weight perfect matching (Fig. 4b). We attribute many of these logical errors in the higher-distance codes to a high-energy impact, which can temporarily impart widespread correlated errors to the system[57]. These events may be identified by spikes in detection event counts[42], and such error mechanisms must be mitigated for scalable quantum error correction to succeed. In this case, there was one such event; after removing it (0.15% of trials), we observe a logical error per cycle of $(1.6 \pm 0.8) \times 10^{-7}$ (see Supplementary Information). The repetition code results demonstrate that low logical error rates are possible in a superconducting system, but finding and mitigating highly correlated errors such as cosmic ray impacts will be an important area of research moving forwards.

## Towards large-scale quantum error correction

To understand how our surface code results project forwards to future devices, we simulate the logical error performance of surface codes ranging from distance-3 to 25, while also scaling the physical error rates shown in Fig. 1c. For efficiency, the simulation considers only Pauli errors. Figure 4c,d illustrates the contours of this parameter space, which has three distinct regions. When the physical error rate is high (for example, the initial runs of our surface code in Fig. 3c), logical error probability increases with increasing system size ($\varepsilon_{d+2} > \varepsilon_d$). On the other hand, low physical error rates show the desired exponential suppression of logical error ($\varepsilon_{d+2} < \varepsilon_d$). This threshold behaviour can be subtle[58], and there exists a crossover regime in which, owing to finite-size effects, increasing system size initially suppresses the logical error per cycle before later increasing it. We believe our experiment lies in this regime.

Although our device is close to threshold, reaching algorithmically relevant logical error rates with manageable resources will require an error-suppression factor $\Lambda_{d/(d+2)} \gg 1$. On the basis of the error budget and simulations in Fig. 4, we estimate that component performance must improve by at least 20% to move below threshold, and substantially improve beyond that to achieve practical scaling. However, these projections rely on simplified models and must be validated experimentally, testing larger code sizes with longer durations to eventually realize the desired logical performance. This work demonstrates the first step in that process, suppressing logical errors by scaling a quantum error-correcting code—the foundation of a fault-tolerant quantum computer.

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

**Google Quantum AI**

Rajeev Acharya[1], Igor Aleiner[1,2], Richard Allen[1], Trond I. Andersen[1], Markus Ansmann[1], Frank Arute[1], Kunal Arya[1], Abraham Asfaw[1], Juan Atalaya[1], Ryan Babbush[1], Dave Bacon[1], Joseph C. Bardin[1,3], Joao Basso[1], Andreas Bengtsson[1], Sergio Boixo[1], Gina Bortoli[1], Alexandre Bourassa[1], Jenna Bovaird[1], Leon Brill[1], Michael Broughton[1], Bob B. Buckley[1], David A. Buell[1], Tim Burger[1], Brian Burkett[1], Nicholas Bushnell[1], Yu Chen[1], Zijun Chen[1], Ben Chiaro[1], Josh Cogan[1], Roberto Collins[1], Paul Conner[1], William Courtney[1], Alexander L. Crook[1], Ben Curtin[1], Dripto M. Debroy[1], Alexander Del Toro Barba[1], Sean Demura[1], Andrew Dunsworth[1], Daniel Eppens[1], Catherine Erickson[1], Lara Faoro[1], Edward Farhi[1], Reza Fatemi[1], Leslie Flores Burgos[1], Ebrahim Forati[1], Austin G. Fowler[1], Brooks Foxen[1], William Giang[1], Craig Gidney[1], Dar Gilboa[1], Marissa Giustina[1], Alejandro Grajales Dau[1], Jonathan A. Gross[1], Steve Habegger[1], Michael C. Hamilton[1,4], Matthew P. Harrigan[1], Sean D. Harrington[1], Oscar Higgott[1], Jeremy Hilton[1], Markus Hoffmann[1], Sabrina Hong[1], Trent Huang[1], Ashley Huff[1], William J. Huggins[1], Lev B. Ioffe[1], Sergei V. Isakov[1], Justin Iveland[1], Evan Jeffrey[1], Zhang Jiang[1], Cody Jones[1], Pavol Juhas[1], Dvir Kafri[1], Kostyantyn Kechedzhi[1], Julian Kelly[1], Tanuj Khattar[1], Mostafa Khezri[1], Mária Kieferová[1,5], Seon Kim[1], Alexei Kitaev[1,6], Paul V. Klimov[1], Andrey R. Klots[1], Alexander N. Korotkov[1,7], Fedor Kostritsa[1], John Mark Kreikebaum[1], David Landhuis[1], Pavel Laptev[1], Kim-Ming Lau[1], Lily Laws[1], Joonho Lee[1], Kenny Lee[1], Brian J. Lester[1], Alexander Lill[1], Wayne Liu[1], Aditya Locharla[1], Erik Lucero[1], Fionn D. Malone[1], Jeffrey Marshall[8,9], Orion Martin[1], Jarrod R. McClean[1], Trevor McCourt[1], Matt McEwen[1,10], Anthony Megrant[1], Bernardo Meurer Costa[1], Xiao Mi[1], Kevin C. Miao[1], Masoud Mohseni[1], Shirin Montazeri[1], Alexis Morvan[1], Emily Mount[1], Wojciech Mruczkiewicz[1], Ofer Naaman[1], Matthew Neeley[1], Charles Neill[1], Ani Nersisyan[1], Hartmut Neven[1✉], Michael Newman[1], Jiun How Ng[1], Anthony Nguyen[1], Murray Nguyen[1], Murphy Yuezhen Niu[1], Thomas E. O'Brien[1], Alex Opremcak[1], John Platt[1], Andre Petukhov[1], Rebecca Potter[1], Leonid P. Pryadko[1,11], Chris Quintana[1], Pedram Roushan[1], Nicholas C. Rubin[1], Negar Saei[1], Daniel Sank[1], Kannan Sankaragomathi[1], Kevin J. Satzinger[1], Henry F. Schurkus[1], Christopher Schuster[1], Michael J. Shearn[1], Aaron Shorter[1], Vladimir Shvarts[1], Jindra Skruzny[1], Vadim Smelyanskiy[1], W. Clarke Smith[1], George Sterling[1], Doug Strain[1], Marco Szalay[1], Alfredo Torres[1], Guifre Vidal[1], Benjamin Villalonga[1], Catherine Vollgraff Heidweiller[1], Theodore White[1], Cheng Xing[1], Z. Jamie Yao[1], Ping Yeh[1], Juhwan Yoo[1], Grayson Young[1], Adam Zalcman[1], Yaxing Zhang[1] & Ningfeng Zhu[1]

[1]Google Research, Mountain View, CA, USA. [2]Department of Physics, Columbia University, New York, NY, USA. [3]Department of Electrical and Computer Engineering, University of Massachusetts, Amherst, MA, USA. [4]Department of Electrical and Computer Engineering, Auburn University, Auburn, AL, USA. [5]Centre for Quantum Computation and Communication Technology, Centre for Quantum Software and Information, Faculty of Engineering and Information Technology, University of Technology Sydney, Sydney, New South Wales, Australia. [6]Department of Physics, Institute for Quantum Information and Matter, and Walter Burke Institute for Theoretical Physics, California Institute of Technology, Pasadena, CA, USA. [7]Department of Electrical and Computer Engineering, University of California, Riverside, CA, USA. [8]USRA Research Institute for Advanced Computer Science, Mountain View, CA, USA. [9]QuAIL, NASA Ames Research Center, Mountain View, CA, USA. [10]Department of Physics, University of California, Santa Barbara, CA, USA. [11]Department of Physics and Astronomy, University of California, Riverside, CA, USA. ✉e-mail: neven@google.com

## Data availability

The data that support the findings of this study are available at https://doi.org/10.5281/zenodo.6804040.

**Acknowledgements** We are grateful to S. Brin, S. Pichai, R. Porat, J. Dean, E. Collins and J. Yagnik for their executive sponsorship of the Google Quantum AI team, and for their continued engagement and support. A portion of this work was performed in the University of California, Santa Barbara Nanofabrication Facility, an open access laboratory. J.M. acknowledges support from the National Aeronautics and Space Administration (NASA) Ames Research Center (NASA-Google SAA 403512), NASA Advanced Supercomputing Division for access to NASA high-performance computing systems, and NASA Academic Mission Services (NNA16BD14C). D.B. is a CIFAR Associate Fellow in the Quantum Information Science Program.

**Author contributions** The Google Quantum AI team conceived and designed the experiment. The theory and experimental teams at Google Quantum AI developed the data analysis, modelling and metrological tools that enabled the experiment, built the system, performed the calibrations and collected the data. The modelling was carried out jointly with collaborators outside Google Quantum AI. All authors wrote and revised the manuscript and the Supplementary Information.

**Competing interests** The authors declare no competing interests.

**Additional information**
**Correspondence and requests for materials** should be addressed to Hartmut Neven.
