## [Peer Review File · Nature]

Manuscript Title: Suppressing quantum errors by scaling a surface code logical qubit

Redactions – Third Party Material

Reviewer Comments & Author Rebuttals

Reviewer Reports on the Initial Version:

Referees' comments:

Referee #1 (Remarks to the Author):

This is an impressive and important piece of work which, for the first time, tries to lower the logical error rate of a qubit encoded in a 'fully-quantum' code by adding more redundancy (going from several distance-3 surface codes to a distance-5 code). The main result is that the logical error rate per cycle of the distance-5 code is estimated to be slightly less than the average of the distance-3 codes, showing that one is in a region of 'component error space' where the surface code is starting to work.

The accomplishments of this paper are 4-fold.

- (1) chip calibration and optimization and run of experiment (for $d=3$, $d=5$ and repetition code)
- (2) applying and optimizing the decoder.
- (3) numerical simulations of the experiment which aim at matching experimental data
- (4) Based on simulations, an estimate of the error budget, and study of the effect of reducing noise in all components.

So, besides detailing and discussing the chip and its optimisation (1), the various ways of decoding (2) which makes this achievement possible, the paper also provides a perspective on the important question of what physical noise suppression it would take to genuinely further suppress the logical error rate (4). In addition, it shows that for a repetition code one can achieve so-called 'algorithmically-relevant error rates' by adding redundancy, but also points out the detrimental effect of cosmic rays (which has been studied before by the team). The paper also shows that quantum simulations (3), varying from density matrix simulations (for $d=3$) to scalable Pauli+Leakage simulations and bare Pauli simulations are able to capture the performance of the device fairly well, in terms of defect rates, logical error rates, and inferred 'edge' error probabilities.

The data analysis, for example, how does one properly estimate the logical error rate per cycle, and how does one compare the $d=3$ codes with the $d=5$ codes (matters on which the claim of achieving an improvement rests) are discussed and treated in good detail.

Despite my overall positive impression of the experimental and numerical accomplishments of this paper, I find the writing and clarity of the paper overall lacking and rather sloppy, in particular if this paper is aiming for the Nature journal.

The paper gives the impression that some sections of it were too hastily written. The fact that this is a large industrial team effort with many contributors may have contributed to this. The consequence is that the paper lacks some cohesion, in terminology, in what information is provided and what is omitted, and how other literature is discussed or explained.

I strongly believe that the authors should improve this before the paper can be published as a solid academic piece of work, i.e. in its current form it is not of sufficient quality. This critique also relates to the fact that the technical data analysis aspects behind the results are not explained in a manner which is very accessible to a reader who has not already read the most recent papers on the subject (often by members of the same team). It is a bit uneven

what data is provided in the Suppl. Material and what information is left out (overall some more basic data could be provided in the Suppl. Material).

Another critique is that I find some of the conclusions of the paper not sufficiently substantiated and some of the observed behavior in the simulation in Fig. 4c and d and also in the repetition code data in Fig. 4b not sufficiently well understood.

I believe that with more work the paper can be substantially improved and the authors should take the time to do so. My list of comments (a wide mix of small and larger issues) is below.

Language throughout the paper:

I find articles missing in a variety of sentences, maybe this is modern parlance but the language often comes across too casually and 'jargony'. I have noted a few concrete cases below.

Notation: Pauli Z (or X, Y) is sometimes italic, sometimes roman, please be consistent.

In a variety of places you write "logical error" as a placeholder for "logical error rate", for example page 1 "logical errors should be reduced ..", "demonstrating reduced logical error..".

In the Suppl. Material there are sections where references to Figs (say Fig. S9) are just written as "found in S9", better to consistently use Fig. S9 etc. This occurs at various places.

In several places in main text and suppl., you leave no space between word and a reference, like[1]. One occurrence of ref -> Ref.

You refer to 'component errors' or 'component Pauli errors'. Even though I think I understand what you mean, it is not standard language, so being explicit about what you mean helps. For example, is cross-talk a component error? A circuit consists of elementary components (the planned gates and operations or idle locations) which all can fail/act differently than intended so I assume that these are component errors. In the caption of Fig. 4 you refer to 'operation errors' which I assume is the same as component error. The short-hand "idle" or "idles" seem rather informal (meaning the circuit location during which a data qubit is idling).

Other typos/sloppy text in main text and Suppl Material:

Various occurrences of "the the"

Linbladian, Gottesmann -> Gottesman.

Typo "hypothesise."

"A higher order effect cause by.."

"per gate and total of n.." -> "per gate and a total of n.."

"that Twirling Approximation...strongly over predicted.." -> that a twirling approximation ..strongly overpredicted...

"of the modeled error channels, (such as dephasing)"

Caption Fig. S20 Pauli+

"To estimate probabilities..." -> "To estimate the probabilities.."

"logical performance of our surface code qubits" -> "logical performance of our surface code qubit"

"Fitting error versus rounds to.." -> "Fitting error rates to extract the logical.."

"developed data analysis.." -> "developed the data analysis.."

Abstract

sentences are awkward and a bit sloppy. Here are some comments:

logical qubits within physical qubits" -> "encoding logical qubits into physical qubits"
"..we report the measurement of logical qubit performance scaling.." -> "we report on logical qubit performance across.."
"we find our.." -> "we find that.."
" per round floor set...": although it is clear what is meant, is this 'floor set' a grammatical construction?
"begins to improve...illuminating the path..": what is begin to improve, did it improve or not? I am not sure in what sense the experiment is 'illuminating a path'

Introduction

"..suppress the operational..physical qubit overhead" -> "physical qubit and temporal overhead". It is known that the surface code leads to a slow-down of computation by a factor d where d is the distance, so this may be good to make clear. On that note, not even the logical error rate per round ϵ is the right performance metric demonstrating that one code is better than another, although we expect that this logical rate per round will decrease exponentially with code distance d , once one is below threshold. Demonstrating such exponential scaling (as you do for the repetition code) is of course the next milestone.

"which traverse the lattice" -> "which traverses the lattice"

Fig. 1 and elsewhere, I am surprised that you do Hadamard gates and no $R_y(\pi/2)$ (and $R_y(-\pi/2)$) gates or should I read Hadamard as $R_Y(\pi/2)$?

Fig. 1c has no vertical axis label (!), no scale and just says "Cumulative distributions of errors", I assume it is fidelity or infidelity here. At the bottom of the caption, would it not be useful to refer to some of the Suppl. Material Section XI where the benchmarking is described in detail?

Caption Fig. 1 on measurement time: it would be good to list (average) time duration of measurement, single-qubit and two-qubit gates in a Table for clarity.

"nearest neighbours except on.." -> nearest neighbours except at.."

T_2 ,CPMG is measured at operation point or at a flux sweet-spot?

Paragraph at the end of Introduction on XZZX code:

"We also remove certain Hadamard..": don't you just move some Hadamard gates around and not remove?

"bias in measurement error": you mean bias in readout error rate, i.e. measure qubit in 0 is more likely to be correctly identified as qubit does not decay during measurement. I don't see this discussed in Suppl. Material Section E, while it prompts you to choose a particular setup, so it is not unimportant.

The XZZX may be less familiar to readers and has been proposed for dealing with biased noise, while here you just care about symmetrisation. One thing which is not clear is how it affects decoding. Usually MWPM on the regular surface code decodes X and Z errors separately. Are you still doing this now but 'reinterpreting an X error' as a Z error (and vice versa) when there was a Hadamard around the CZ gate in Fig. S12? I believe this is correct as you write in Section V in the main text about decomposing into disjoint error graphs (however I could not find this discussion in the Suppl Material on decoding).

If so, I am also puzzled about the statement that Y errors cause more than 2 detection events in Section XIV. C in Suppl. Material. This is true for the complete error graph, but if we split the error graph into 2 parts and do matching separately, such Y errors correspond to proper edges.

This also relates to another issue about the description of the decoder: you refer to the "correlated MWPM method" in the Suppl. Material: "uses a variant of the two-pass correlation strategy detailed in [63]". What correlations these are (Y errors being X and Z errors) and how correlations are used is never really discussed in Section XII. B. Please give some high-level idea about the correlated MWPM (after the BP algorithm).

Figure 2:

Figure is quite dense, there is barely space for Figure h.

"...that manifest in detection pairs" -> "...that manifest in detection pairs, except at the boundary".

The information in Fig 2a is hard to parse, in particular what are the pairs when they are timelike or spacelike separated (purple circles and blue circles).

Caption Fig. 2: "...no preceding data idling": is this really the reason and could it not be because of measurement errors. A measurement error causes defects in the current and the following cycle. A measurement error in that cycle can give a defect, but at $t=1$ there are measurement errors that happen at $t = 1$ as well as the defects coming from $t=0$.

"The average detection..roughly 5 cycles [43]": is this the number of cycles for the qubits to reach their leakage steady state? Could be discussed in more detail.

For leakage removal the authors cite [42,47-49], but [42] and [49] only considers removing measure qubit leakage while from the Suppl. Material it is clear that data qubit leakage is important. Other work such as [78] proposes data qubit leakage mitigation, while <https://doi.org/10.1038/s41534-020-00330-w> studies aspects of leakage and leakage detection more generally.

On Fig. 2h and matching with numerics.

First, the " p_{ij} method" is not widely known and you should define the "correlation p_{ij} " somewhere in the main text (and in the Suppl. Material Section which discusses this) as you do in [42]. More importantly, it is not wholly clear why this correlation p_{ij} is a relevant quantity: this is because these are approximately the error probabilities which lead to pairs of defects. Also this method was first executed in Spitz et al.

<https://onlinelibrary.wiley.com/doi/10.1002/qute.201800012> (which is not cited).

What are you doing with edges which only lead to one defect at the boundary. You are not plotting these probabilities anywhere or mentioning them, is your simulation also estimating these according to what you extract from the experimental data?

Also, in caption of Fig. 2h: you mention diagonal edges here, but these are called spacetime-like edges (ST) elsewhere and there is a class of unexpected spacetime like edges ST'. It would be better to stick with 1 naming convention here.

On your claim that you match performance with simulation: in the main text in Section IV, the description of Pauli, and Pauli+ is quite brief and hard to understand what is really done, e.g. what does it mean to include 'coherence information'?

It is also worthwhile to dwell a bit more in the Suppl. Material on the disadvantages/weaknesses of your error model. For example, you add a depolarizing channel in order to match the observed error rate in RB/XEB that have no physical explanation (in Section XIV.A.5.1 Depolarizing errors).

Also, it seems that in simulation the leakage rates were uniformly picked while some qubits are clearly leaking more than others. Your crosstalk model used in simulation does not seem to be discussed (and you refer to later work)

Another point to bring up here is that your Pauli+ simulation, although clearly an interesting and useful idea to better model the experimental data, cannot capture the fact of course that leakage is state-dependent (i.e in a CZ $|11\rangle$ can leak to $|02\rangle$, but not the other states, and heating is expect to lead from $|1\rangle$ to $|2\rangle$ less indirectly from $|0\rangle$ to $|2\rangle$). You model this by assuming the qubit is in a random state and reducing the leakage rate correspondingly, but it is a feature that one cannot model easily.

Section IV

"Measurement and reset errors...timelike pair": also errors on measure qubits during the cycle can lead to such timelike pair. Similarly, for the spacelike pair there are contris from single and two-qubit errors. If you want to focus on the leading errors in this text, or just naming some example, please make this clear.

Section IV

You write “To estimate probabilities of detection event pairs..an appropriately normalized correlation p_{ij} ..” and you refer to [42], but you should also cite the Spitz et al. paper (see comment above). Furthermore, the sentence is completely unclear, what is appropriately normalized, you don’t estimate the probability for a detection pair, you estimate the probability for a single error which leads to the detection pair.

Section V

Sentence “and each hyperedge is assigned its corresponding error mechanism probability” sounds very complex (i.e. “error mechanism probability”). The important point of this model is that it is a stochastic model and cannot capture coherent errors (amplitude versus probabilities) and that it assumes that errors at circuit locations occur independently, and this should be added.

For each hyperedge you estimate an error probability. I am not sure that the issue of hyperedges “generalized p_{ij} ” is described in a satisfactory way in the paper, the brief section Suppl. Section XVI. C is devoted to it and only sketches in a perfunctory manner how this works. The way it influences the results in the paper seems to be that it gives a better estimate for p_{ij} (avoiding negative probs) and thus the weights $-\log(p_{ij})$ in the decoder, like in [50]. I note that how the weights depend on p_{ij} is not even mentioned in Section XII on decoding. Trying to understand anything from Suppl. Section XVI is frustrating and does not help the quality of the paper so I would suggest that either the authors expand on describing this method well (if it is important in optimizing the decoder which is not clear from the text), or defer a solid description to another paper.

You write that the tensor network decoder depends on the “hypergraph error priors” (section V main text): are these ‘priors’ what comes out of the hyperedge analysis and how do they influence the tensor network decoder?

Overall, it seems that Section XVI should be better integrated to Section XII on decoding where priors are also mentioned (or refer in Section XVI for details covered by Section XII) and the ‘extended p_{ij} ’ method is also discussed.

Section V

“These graphs are decoded efficiently...minimum weight perfect matching”. From the Suppl. it is clear that the actual procedure is more involved, “correlated MWPM with some BP before this”, so this does not capture what is actually done. Without giving all details, the idea could be better presented.

Section V

You define the logical fidelity $F=1-2 P_L$. For large numbers of cycles, one expects that the logical error probability $P_L=1/2$, essentially randomizing the logical qubit (see Fig. S14), and then the fidelity $F=0$, while a randomized qubit has fidelity $1/2$ with any pure state. So why not use $F=1-P_L$ which is the proper fidelity expression, assuming a logical depolarizing channel which maps $\rho \rightarrow (1-2p_L)\rho + p_L I$.

This ties in with the modelling in Suppl. Section XIII (see other comments below on that section), where you derive Eq. (2) and Eq.(2) does not follow using $F=1-P_L$ (since you go to fidelity= $1/2$ instead of an exponential decay to 0). Of course you are free to define “a fidelity” and just fit it with an exponentially decaying curve which goes to 0, but terminology may be confusing.

ϵ should be defined in main text instead of in Eq. (2) in this Suppl. Section, since it is the important parameter.

Section V High-energy event discussion:

You don’t post-select on leakage or high-energy events, but it may still be interesting to know what the error rates would be if you did so. Related to this you write “These events may be identified by spikes in detection event counts” but you don’t specify in the Suppl. over what time window of runs you average the detection probability and throw out these runs when they are above some threshold, i.e you don’t specify the procedure for tossing out those data. Concerning high-energy event issues, it is also not fully clear to me that this is an issue that occurs for other superconducting transmon teams (say the IBM group who has many

processors in the cloud in continuous operation, so should know about this if present), or whether it can be mitigated by better shielding.

Section VI

On Fig 4: how can the fact that this only affects the logical error rate of the repetition code at very high distance be understood? I presume that the smaller distance codes are obtained from taking subsamples from the larger repetition code? But then a high-energy event, affecting say all T_1 , also affects the smaller codes, increasing the logical error rates there as well, so how do you understand this?

Perhaps this indirectly relates to understanding the curves in Fig.4d (for the surface code), namely in the cross-over region, the logical error rate seems to first go down and then sweep up again (see next comment).

Figure 4d:

It is not clear that $s=0.9$ really is below threshold - is that clearly seen from the $d=25$ data point- it seems to be going flat, but it could also be a matter of scale. I think it'd be better to show.

More importantly, why would the logical error rate per cycle in the 'cross-over regime' first go down and then go up again as the curve seems to show? In the text you refer to finite-size effects, but you don't analyze it properly. It is useful to write an expected expression for ϵ_d and see how this matches your data. One would expect that ϵ_d should grow exponentially in d above threshold while decreasing exponentially in d below threshold, while it can have some prefactor, some function in d , which gives finite size effects. If this prefactor is monotonically increasing in d , I don't think one can get the scaling you have found (i.e. dip and then increase). This is a bit puzzling since one expects that the prefactor consists of some combinatorial effect (scaling as $\text{poly}(d)$) divided by d since ϵ_d is the error rate per cycle. One reason that there is first a dip in logical error rate and then an increase as a function of d is that the decoder for large d is not decoding up to the distance really (because computational cost will surely increase with d). This would be an important problem. Hence I believe that the authors should work on understanding/explaining this data in a more convincing manner (or put error bars on the curves if the shapes of the curves are misleading).

Could you provide a definition of the error suppression factor Λ ? (you refer/introduce it in Section VII but it is hard to find). In Section V you give it with a subscript $\Lambda_{\{d/d+2\}}$ and you mention it in Section VII without subscript (so any asymptotic large d version of it?). In Section XI.G there is also a $\Lambda_{\{35\}}$ typo and in that section it also mentions Λ without subscript.

Section VII

You write "For efficiency, the simulation considers only Pauli errors" So why not Pauli+, is this not efficient enough?

Section VII

One of your main conclusion is that your experiment "lies in a cross-over region...". As I mentioned above, if this is one of your conclusion it needs to be substantiated with a better analysis of the finite-size effect behavior. This is also important for interpreting what is seen in future experiments.

You write "Based on the error budget... component performance must improve by at least 20%..": does this somehow obviously follow from the curves for differing s ? Could you clarify this? Also, is it just device performance or also decoder performance which needs to improve?

General comment on information about temporal execution of the experiments:

I believe that it would benefit the community if more information is provided about timescales of fluctuations in parameters and over what period of time data has been taken.

In Section XI.B you write about random fluctuations due to TLS, but no typical time-scales or information about how frequent one re-optimizes versus the overall time for data acquisition. In Section XI.G.3 you provide some details related to ordering of experiments, but over what period of time data are taken and how much re-calibration is done in between is not made

clear.

Suppl. Information:

Sect XI

Section XI.B. shouldn't $f_{10/21}$ be $f_{11/20}$ as that is the avoided crossing which the CZ uses. "distance-5 objective": what do you take as objective here? The performance of parity checks?

Section XI.C

Are you determining L1 of each single qubit gate and/or two-qubit gates as well? For single-qubit gate you write a whole section XI.F on reducing microwave cross-talk which induces leakage, but you never give what you measure as L1. Similar question on L1 during readout (which contributes to readout error in Fig S9).

Do you measure other parameters like the leakage swap or the phases leaked qubits impart on computational qubits if those are measured (phases are mentioned in Section XIV.A.3). If these parameters are used in the error model in simulations, they should be provided.

Fig. S7

The fig refers to T_2 but you used T_2^{CPMG} before, so denote by the same symbol? Quite a lot of qubits are breaking the $T_2=2T_1$ limit. I guess this can always happen if there are fluctuations and recalibrations between the times these datasets were taken, but I think it is useful if the authors comment on how they understand this.

Fig S8 and S9

Please provide definition of coupling efficiency and measurement efficiency. Define readout-resonator coupling in % .

"CZ...as measured in the surface code layers" is not clear. Figures occur quite late after associated text, so move up.

You don't give reset error probability and describe what reset does: does reset include reducing leaked states to computational basis states?

Section XI.E

"swapping between neighboring qubits", does this imply all transitions among the 4-level systems, i.e. 01-10 and 12-21 and 21-03 and 12-30, more or less?

Section XI.F.

Define/introduce receiver and source qubit before using it. Are source and receiver qubits neighbors or one only needs to have f_{01} close to f_{21} .

"We apply a...for V'..." -> "We apply a...with V.."

Section XI.G

"We present the surface.." -> "We have presented.."

"..which minimized the detection rate in parallel.." : do you mean average detection rate (over time or over measure qubits?) or the detection rate/probability for each measure qubit. Be more specific.

Section XI.G choice of $d=3$ grids

"excess phase errors": Excess compared to what and for what reason? Did T_2 go down by what?

"our models": what simulation model are you referring to here?

"outlier qubits" -> boundary qubits

Caption Fig. 11

You refer to 'moments' and 'single moment' (which may relate "Moments" in Cirq), but steps is

more standard terminology.
“Note although..” -> “Note that although..”

Section XI.G or elsewhere:

Somewhere it can be mentioned how to get the data for $d=3$ from $d=5$, i.e. set the other qubits to 0 (or +) so that weight-4 parity checks only depend on the data qubits of the $d=3$ code. This can lead to extra error at the boundary if the non-participating qubit needs to be in +. Or are you doing a different experiment with weight-2 checks for the $d=3$ cases? Please clarify, perhaps in main text. This comment also applies to the repetition code.

Section XII Decoding

“To account for this...”: do you mean a Pauli-twirled version of T_1/T_2 , but what duration is picked, I presume the duration of the location which represents the edge.

Nothing is mentioned about boundary edges here, i.e. error events which lead to a single detection event and how well and how these are estimated by this method.

Typo “averaged over round”

Section XII.C

“error hypergraph is decomposed into..”: more details would be useful.

You write “Practically, this decoder...”: the statement does not refer to what size code you are decoding, so I assume you believe it works for any d ? (but this would require parallel execution). If not specify what d .

Section XIII

Please say what r is (number of rounds).

Please write what N is

You could provide some references where modeling is done differently.

The paragraphs on quantification of out of model errors are interesting but hard to follow, it just reads as a short hand description of some numerical procedures which a reader cannot verify nor follow for correctness really. Not sure how easy it is to improve this.

Section XIV

“depolarizing channels each device operation”. Experimentally characterized fidelities: could you be clear what data of what Section is used and refer to it.

On brute force, quantum trajectory simulations: these are done in some other papers as well, in particular for modeling leakage and logical performance for the surface code, see e.g. [84].

Section XIV.A.1.

H_{idle} is not defined, just give an expression/description for it.

Section XIV.A.3

“in characterization” -> “in the characterization”. What is meant by isolated CZ gates? Just CZ gates? Or CZ gates not run in parallel when other gates are run?

“The qubit ordering above matches the f_{10} transition frequencies...”: I don't follow this sentence, how does qubit ordering in an equation influence what qubit leaks.

Section XIV.A.4

You mention 2 sources of cross-talk: inactive couplers and ‘diagonal capacitive couplings between next-nearest-neighbor qubits’. I don't know what is ‘diagonal’ about capacitive couplings. Aren't these all treated/leading to ZZ interactions?

Section XIV.A.5 “fidelity decay curves are converted to an inferred Pauli error”. How? You get an estimated Pauli error from physical modeling and then you have to add some extra depol noise to get the fidelity, but doing this is not unambiguous.

Section XIV.B. Simulator Details

Section XIV.B.2.

“A perfect rest to a single-qubit device $|0\rangle$ state”: in this section you choose to refer to the qudits as devices, this is ok as a choice but could be mentioned at the start.

In eq. (15) why don't the Pauli operators act on qubits in D (they transition down and now get a random Pauli on them.)

I find the pseudocode description not superclear. When you introduce \hat{G}_i (first with hats, then without..), include $G_{i[j]}$ as the j th entry of this Pauli being I,X,Y or Z.

Section XIV.B.3

Good to be concrete in how you get the twirled channel, but some mention on how to extract Kraus operators from the Choi state which characterizes the channel could make this section be more embedded in standard QI theory (instead of reading like pseudocode)

QMC simulation, does this stand for Quantum Monte Carlo? Isn't this just full density matrix or 'brute force/quantum trajectory simulation that you mention earlier, QMC suggests other things to me and the abbreviation is not even written out when it first occurs/is described in XIV.B.1.

Fig S21

Would be interesting to see the same plots but for Pauli+ and experimental edges. I think this is particularly important since the Pauli+ are used for error budgeting and “physical understanding”.

Section XV

Here you refer to ϵ_L (L stands for logical now?) while elsewhere you use the subscript ϵ_3 and ϵ_5 .

Why is the half-operation point a relevant point to evaluate sensitivities at?

Section XV.A paragraph “In particular, the SQ gate...” is a repeat of information mentioned elsewhere.

Abbreviation Dq in Figs S22/23 not previously used.

Section XVI.A : perhaps you can point out where hyperedges due to Y errors occur in Fig. S24 (as you mention them earlier..)

Section XVI.B time-like edges. You mention that leakage and CZ also contribute to T edges, but don't say anything about the mechanism.

Section XVI.B space-like edges: clarify what an SX edge is.

“and at some round t” -> “at some round t”

“also contributes..” -> “also contribute”

“T1 qubit decay” -> “T₁ qubit decay”

Fig. S29 leakage in D qubits, what are D qubits?

XVII

“use error probability..” -> “use an error probability”

You mention that most likely underestimate leakage in simulation (I presume you mean leakage of qubit 2₄ and 2₆): do you any idea that this is due to heating or due to CZ gates, or yet some other mechanism.

“solving numerically this nonlinear systems..” -> “solving numerically this nonlinear system”

Section XVII

Perhaps use vector notation in p_{ext} and w for clarity.

“use error probability..” -> “use an error probability”

Section XVIII

The table is representative although I find your description of bosonic error correction (“embedding QEC into the physical qubits”) a bit misleading. Maybe better to say “embedding QEC into the definition of a physical qubit as a subspace of an oscillator”

Referee #2 (Remarks to the Author):

General comments

The authors of “Suppressing quantum errors by scaling a surface code logical qubit” realize a distance-5 surface code logical qubit and demonstrate that its logical error rate is slightly lower than the average of the logical error rates of four distance-3 logical qubits implemented on the same device. Accurate modeling of the experiments, including leakage errors, allows the authors to understand the main error sources limiting the logical performance. Furthermore, the authors run a distance-25 repetition code finding no obstacles for achieving algorithmically relevant error rates in the future except for the impact of high-energy radiation.

The authors emphasize the modest observed performance gain from $d=3$ to $d=5$ and underpin it with a thorough statistical analysis in supplementary information. In my opinion, the work would not lose in importance, if e.g. only equal logical performance was demonstrated. In fact, the number of qubits is almost tripled from $d=3$ to $d=5$ and so is the number of operations executed in parallel. Maintaining performance in such a setting is a major experimental achievement and raises hope that fault-tolerant quantum computers based on a surface code architecture can be built in the future. Prior to this work, only distance-3 surface codes had been experimentally realized using superconducting circuits (Refs. [22] and [24] of the paper). Another strength of the paper is the thorough modeling of the experiment, including not-so-obvious-to-model but important error sources such as leakage and crosstalk. Understanding them in detail and determining the main error contributions to the logical error rate measured is indeed very important to improve device performance in the future.

The main body of the text is accessible and clearly written and the conclusions drawn appear to be correct. The supplementary material supports the claims of the paper and provides most of the details required for an in-depth understanding of the experiment and the simulations. However, providing more detail on the device architecture and its implementation would be essential for any other party to be able to reproduce the results presented in the manuscript.

Our main criticism concerns the following points:

1. We find that the explanation and motivation for using the ZXXZ surface code is not sufficiently detailed. The authors motivate it with symmetrizing X- and Z-error rates, but it is not clear to us why this is desired. On the contrary, we would find it interesting to distinguish between the impact of data qubit energy relaxation errors and dephasing errors on the logical performance. The authors do not comment at all on the more conceptual implications of using the ZXXZ surface code: Is it expected to perform better than the standard surface code on their device / have the authors run the standard surface code on their device? What are the differences regarding the decoder in the view of the material presented in Ref. [46] by Ataides et al.? Also the logical operators of the ZXXZ surface code are not presented. We feel that a separate section in the supplementary information on the ZXXZ surface code would be very helpful. We understand that focusing the discussion on the standard surface code in the main text is more pedagogical, but we would recommend making connections to the actual code implemented whenever possible.

2. Little details are provided for the repetition code experiment. The data set presented suffered from a single high-energy event: How representative is this data set? How long was the data acquisition time and what is the number of high-energy impact events expected during that time based on the authors' previous study? Generally, we think the repetition code does add only little to the already nice achievements of the distance scaling experiment. The general observations presented in the two paragraphs of this section are rather short and have been stated in related previous work by the same team published in the same journal. Also, the impact of high-energy disturbances is not discussed in much detail here, while it is being discussed by the same team in other papers. If needed for space constraints, the whole section could be moved to the supplementary material, for example.

3. The readout architecture of the device is not sufficiently documented (Purcell filters, multiplexing, use of near-quantum-limited amplifiers etc.). Furthermore, in the experimental details section of supplementary information, some standard parameters such as single-qubit and two-qubit gate durations are missing, and some of the parameters are stated in a non-standard way without defining them (effective qubit-qubit coupling efficiency, resonator-qubit coupling). For detailed comments, please see below.

Overall, this is a great result, and we shall recommend publication in Nature if the points raised above and the points detailed below are addressed carefully.

Detailed comments on main text

Abstract: "... highlight the biggest challenges for future systems ...". Please specify what these are from your perspective. Otherwise, the statement is assertive while being very generic.

Abstract: "... the first experimental demonstration ...". My understanding is that claims of novelty are usually not allowed by editorial standards in most journals, even if correct.

Section I, beginning of paragraph 1: The references are quoted in a very generic form. It would be useful to the reader to be more precise about what is covered in these references. Some of those present work that is very closely related to the current work, but the authors do not point this out. Others are covering different approaches to QEC. A bit more room for the discussion of the references would be helpful for the reader.

Section II, paragraph 2: Different terms are used for qubits which in the original literature were called ancilla qubits. We think it would be useful to help converge on a common terminology. The authors may want to argue for a suitable convention and help using it consistently. Terms currently used include ancilla, helper, measure, auxiliary qubits and possibly others.

Section II, paragraph 3: It is pointed out that logical memory operation is most important. Equally important however, is the execution of gates between logical qubits. In that context, it would be adequate to also refer to D. Litinski, Quantum 3, 128 (2019) when citing Refs. [37, 38] for lattice surgery.

Figure 1c: The qubit idle errors marked DD seem to be significant but are discussed very little in the main text. Expanding a bit would help the reader appreciating the relevance of those errors.

Figure 1, caption b: "... focusing on one data qubit (gold) ...". Stating more explicitly that the focus is on one data qubit (labeled with ψ) interacting with 4 measure qubits and one measure qubit (labeled with 0) interacting with four data qubits would render the statement more explicit.

Section II, last paragraph: The ZXXZ variant should be discussed in more detail, see general comments as well. Also "This ensures that we do not preferentially measure even parities in the first few rounds of the code, which could artificially lower logical error rates due to bias in measurement error." With respect to this statement, please refer explicitly to the experimental data in the supplementary figure S18 which demonstrates this bias and its effect on the logical error rates, otherwise this may be easily missed or the statement may appear poorly justified.

End of section III.: "We begin with a Pauli simulation based on the component error information in Fig. 1c in a probabilistic Clifford simulator, then incorporate coherence information, transitions to leaked states, and crosstalk errors ...": We are surprised that the authors mention coherence information here. We assumed that the measured coherence times of the qubits are also used in the standard Pauli error model to model idle operations. Can the authors please clarify?

Figure 2, caption a: "... diagonal pair ...". In the literature and in the author's own supplement they refer to these errors as space-time-like errors, which seems to be more commonly used in literature. Thus, we suggest to use this terminology here as well.

Section IV, last paragraph: "We expect that mitigating leakage and stray interactions will become increasingly important as error rates decrease." Can the authors please expand on what they mean by stray interactions. This would help readers to understand the main text.

Section V, 2nd paragraph on page 5: "...leakage accumulation may cause distance-5 performance to degrade faster than distance-3 as logical error probability approaches 50%." Why is that? Is this due to the on average higher qubit-connectivity of the d=5 code? Do the authors have data for logical error probabilities approaching 50%?

Section VI, paragraph 1: "... new dominant error mechanisms may arise ..." Please specify what type of errors you have in mind to be more precise. Also, "... 1D version of the surface code ..." We find referring to a repetition code in this way misleading. Please state that this is the repetition code. The repetition code is essentially attempting to correct errors in a conventional, classical approach.

Section VI. A reference to Figure 4 b containing the relevant data is missing from section VI. Please add so that the panel will not be missed when reading the paper.

Fig. 3c and corresponding text: The authors refer to chronologically improved performance. Is the data taken on the same device and what is the max. time span between data points?

Fig. 2a: Consider moving the red Z-error label a bit to the right to not overlap with the depiction of the surface code parity grid.

Ref. [48] and [59] (McEwen et al., Removing ...) appear to be identical.

Detailed comments on Supplementary Information

XI. A.: "... with XY-4 phase cycling": Reference is missing.

XI. A.: "... the specific dynamical decoupling sequence is chosen on a per-qubit basis in order to ...": Can the authors elaborate on the adjustable parameters in their dynamical decoupling sequences?

XI. C. and Fig. S7: The single-qubit gate duration is not stated.

XI. D.: "CZ gates in the sycamore architecture arise from a state-selective dispersive shift ...": We find the term dispersive shift misleading here since the deployed CZ gates exploit a resonant interaction as far as we understood, not a dispersive one.

XI. D. and Fig. S8: The two-qubit gate duration is not stated. Furthermore, the coupling efficiency plotted in panel (b) is not defined. We assume it is defined with respect to the minimum achievable gate duration. Please clarify.

XI. E. and Fig. S9: Resonator-qubit coupling (panel b) is not defined. We assume this is the ratio between the dispersive coupling strength and the coupling strength if qubit and resonator were on resonance. Please clarify. Furthermore, it would be useful if the authors indicated not only the readout power in dBm in panel g but also the intra-resonator photon number or the conversion factor between the two.

XI. G. 1.: “We can divide the corrections into ten groupings of compatible corrections for any distance of surface code, meaning that the corrections impact different detectors and can therefore be optimized simultaneously... Since the number of groupings is constant for any distance of surface code we expect this method to scale well as we increase the sizes of our system.”: I find these statements hard to follow. Can the authors please explain in more detail?

XII. A. 3rd paragraph: “To avoid this issue, we decode an even subset of experimental trials by computing p_{ij} on the odd subset, and vice versa...” : The authors mentioned earlier that they use a 25 cycle data set taken at the beginning for training the decoder. Was that data set thus not used to set the weights of the decoder? Please clarify.

XII. A. 4th paragraph: “We perform an abbreviated (due to the computational cost of the tensor network) distance-3 decoding experiment using 5, 9, and 13 rounds with this method.” We appreciate that the authors performed such checks. In view of their comment we are curious about the computational cost of the tensor network decoding for $d=3$ and $d=5$. Can the authors give an idea about the computational cost (resources and time), potentially in the tensor network decoding section below?

XIII. 2nd paragraph: “... discarding the first round as it decreases the quality of our fits due to the unique nature of its errors, which disproportionately favor distance-5.”: Do the authors understand why the $d=5$ experiment is favored more than the $d=3$ experiment when including the first round?

XIII. end of 2nd paragraph: “When combining data from multiple experiments, such as when considering data from multiple logical bases, this process is done on each dataset separately, and the final ϵ 's are averaged.”: What about the five different basis states for each logical basis, are they fitted separately or jointly?

XIII. last paragraph: “As a check that this can appropriately account for additional errors, we also fit an extended model in which the residuals in our fits come primarily from fluctuations in the logical error per cycle ϵ itself instead of binomial sampling noise.” To allow us to judge this extended model in comparison to the binomial model, it would be helpful if the authors provided more details on the extended model.

XIV. A. 3. “Since our CZ gates are diabatic (implemented as two complete Rabi swaps)” : We understood that the CZ gate used in this work is the one from Foxen et al. (2020), is this correct? Or why do the authors talk about two complete Rabi swaps rather than a single continuous round-trip in the 11-02 manifold?

XIV. A. 3. Last paragraph: “In order to accurately capture the above processes, we numerically calibrated then simulated the CZ gate matching the experimental qubit and coupler control schedules.”: Was the numerical calibration of the gate in the simulation done using ideal waveforms or were potential waveform distortions included?

XIV. A. 4. “First, the inactive couplers (i.e. those not implementing a CZ gate) used to mediate the qubit-qubit couplings at idle are unable to cancel all effective couplings over all fixed excitation

subspaces”: Can the authors state the dominant residual couplings?

XIV. A. 5. “While the models above account for a majority of errors in the experiment, they typically under-predict the experimentally calibrated decay of fidelity.” I would add something like “... in benchmarking experiments” to avoid confusion with the decay of the logical fidelity. Furthermore, can the authors comment on the origin of the observed discrepancy? Is it due to control errors, additional dephasing upon tuning the qubits in frequency during two-qubit gates, etc.?

XIV. B. 2.: To avoid confusion, it would be good if the authors stated at the beginning that they refer to the simulated qubit systems as devices throughout this section.

XIV. B. 2. 4th paragraph: “As a result, the full Hilbert space is separated into mutually incoherent leakage subspaces given by Kronecker products, ...” Why is it called leakage subspaces if the subspace can also be the computational subspace?

XIV. B. 2. End of 4th paragraph: “As detailed below, approximating quantum error channels in terms of GP channels is premised on the assumption that the input is a highly entangled stabilizer state, ...” It is not clear to me why general Pauli channels should be premised on stabilizer states only. Can the authors please clarify?

XIV. B. 2. End of 5th paragraph: “Initial state of the qubits in U is traced over, while the final state for each qubit in D is a randomly selected $|0\rangle$ or $|1\rangle$ with equal probability.” Doesn’t one expect that the final state for qubits in D is more likely to be $|1\rangle$ than $|0\rangle$, simply because the leakage states first decays to $|1\rangle$?

XIV. B. 1. 3rd paragraph: “This means that no superposition is ever formed between the computational and leakage subspaces of any qubit, and we therefore are never required to keep more than two levels per qubit in our state vector.” This seems like an important assumption and simplification. Can the authors comment on the consequence for the accuracy of the simulations?

Fig. S19, caption: “... so that its standard deviation is bounded from above by $1/\sqrt{4N} \approx 0.11\%$ ”: I didn’t immediately realize that the factor four comes from assuming a binomial distribution for epsilon. Please consider adding this explanation.

XV. The authors state that the sensitivity analysis with respect to epsilon is distinct from the $1/\Lambda$ analysis around Fig. 4a. I find that the conclusion in this section is missing and that it would be helpful to better work out the relation to the error budget presented in Fig. 4a of the main

text.

XV. A. “Finally, another imperfection in the device is the presence of unwanted interactions (crosstalk) between simultaneous gates (either single qubit idle or CZ), ...”: What about crosstalk due to simultaneous single-qubit gates?

Next sentence: “This error channel is modelled individually for all elements and leads to CZ errors with a Pauli error probability (averaged over all CZs) of ..., which is roughly 15 % of the observed total CZ error”:

1. The authors say that crosstalk for single qubit idling is also considered but only mention CZs gates here. Please clarify.
2. I understand that the crosstalk error stated comes from the simulations described in XIV. A. 4. Is this calculated value compatible with XEB fidelities for CZ gates measured individually and in parallel?

XV. B. and Fig. S22: We notice that the components with largest absolute error rates have lowest sensitivity. Is this expected? Wouldn't it make more sense to look at the sensitivity with respect to relative changes in error rates, i.e. use $\frac{\Delta p_{\text{error}}}{p_{\text{error}}}$ instead of Δp_{error} ? At least an experimentalist rather thinks about gate error improvements on a relative scale.

Fig. S24: The pixels in each of the cells have different widths and different heights, which makes it harder to interpret when zooming in.

XVI. B. 2. 1st paragraph: “The number 0.043 written in the middle tile (5,5) is the average (specifically, the median) of the p_{ij} probabilities of all SX edges between these two X measure qubits.”: Do the authors mean the average over cycles / time? If so, this could be said simpler.

XVII 2nd to last paragraph: “we subtract the contribution of 9.5×10^{-4} from crosstalk and the contribution of $1.25 \times 2 \times 10^{-4}$ from CZ leakage.”: It is not obvious to me where the factor of 1.25 in the leakage contribution comes from. Can the authors please clarify?

Table IV: Ref. [24] is from year 2022.

Referee #3 (Remarks to the Author):

In this work the authors have demonstrated suppression of quantum errors by scaling up the surface code from distance 3 to distance 5. Although the actual degree of suppression is small and breakeven has not yet been demonstrated, this work is a landmark in the field and, and I

recommend it for publication. I particularly appreciate the balanced approach taken by the authors in writing the manuscript, wherein they have presented detailed error budgeting and identified necessary areas of improvement in the experiment.

I do have some comments for the authors to consider below.

1. A lot of times some very technical points are mentioned in the main text without reference to the relevant section in supplementary material. Even adding a 1 sentence intuitive explanations for these concepts will make it much more accessible to Nature's audience. One example is the statement "to avoid time-boundary effects that are advantageous to the distance-5 code". Only QEC experts (a smaller subset of QI community) will be familiar with the concept of time-boundary. A short 1-line explanation or reference will be useful here. Another example is " This ensures that we do not preferentially measure even parities in the first few rounds of the code, which could artificially lower logical error rates due to bias in measurement error." More explanation of "bias in measurement error" is provided in the supplementary material although not explicitly cited at this point.

2. The authors note that the leakage gets slightly worse when we move from distance 3 to distance 5 codes. They attribute this to the fact that more gates etc are being executed simultaneously in the dist-5 code. Can the authors say more about how they expect leakage to scale as the system size gets bigger than dist 5? Would it possible to mitigate stray interactions at the hardware level or is the next step to deal this with the error-correction architecture?

3. It would be useful to have a bit more details about the assumptions behind the simulations in Fig.

4d. As far as I understand, that figure ignores leakage. It would be nice to know how leakage will affect that plot. Or even how leakage must scale for that plot to hold.

This is an impressive and important piece of work which, for the first time, tries to lower the logical error rate of a qubit encoded in a 'fully-quantum' code by adding more redundancy (going from several distance-3 surface codes to a distance-5 code). The main result is that the logical error rate per cycle of the distance-5 code is estimated to be slightly less than the average of the distance-3 codes, showing that one is in a region of 'component error space' where the surface code is starting to work.

The accomplishments of this paper are 4-fold.

*(1) chip calibration and optimization and run of experiment (for $d=3$, $d=5$ and repetition code)
(2) applying and optimizing the decoder.
(3) numerical simulations of the experiment which aim at matching experimental data (4) Based on simulations, an estimate of the error budget, and study of the effect of reducing noise in all components.*

So, besides detailing and discussing the chip and its optimisation (1), the various ways of decoding (2) which makes this achievement possible, the paper also provides a perspective on the important question of what physical noise suppression it would take to genuinely further suppress the logical error rate (4). In addition, it shows that for a repetition code one can achieve so-called 'algorithmically-relevant error rates' by adding redundancy, but also points out the detrimental effect of cosmic rays (which has been studied before by the team).

The paper also shows that quantum simulations (3), varying from density matrix simulations (for $d=3$) to scalable Pauli+Leakage simulations and bare Pauli simulations are able to capture the performance of the device fairly well, in terms of defect rates, logical error rates, and inferred 'edge' error probabilities.

The data analysis, for example, how does one properly estimate the logical error rate per cycle, and how does one compare the $d=3$ codes with the $d=5$ codes (matters on which the claim of achieving an improvement rests) are discussed and treated in good detail.

We appreciate the referee's clear summary of our main results and identification of our work as impressive and important.

Despite my overall positive impression of the experimental and numerical accomplishments of this paper, I find the writing and clarity of the paper overall lacking and rather sloppy, in particular if this paper is aiming for the Nature journal.

The paper gives the impression that some sections of it were too hastily written. The fact that this is a large industrial team effort with many contributors may have contributed to this. The consequence is that the paper lacks some cohesion, in terminology, in what information is provided and what is omitted, and how other literature is discussed or explained.

I strongly believe that the authors should improve this before the paper can be published as a solid academic piece of work, i.e. in its current form it is not of sufficient quality. This critique also relates to the fact that the technical data analysis aspects behind the results are not explained in a manner which is very accessible to a reader who has not already read the most recent papers on the subject (often by members of the same team). It

is a bit uneven what data is provided in the Suppl. Material and what information is left out (overall some more basic data could be provided in the Suppl. Material).

We agree that there is room for improvement in the presentation, primarily in the supplementary information, and we are making improvements in light of the referee's specific comments below.

Another critique is that I find some of the conclusions of the paper not sufficiently substantiated and some of the observed behavior in the simulation in Fig. 4c and d and also in the repetition code data in Fig. 4b not sufficiently well understood.

We appreciate the critical look from the referee and hope to improve it based on their constructive remarks. We believe the conclusions are substantiated and address the referee's specific comments on this subject below. We also believe the discussion of Fig. 4b-d are sufficient and address the referee's specific comments below.

I believe that with more work the paper can be substantially improved and the authors should take the time to do so. My list of comments (a wide mix of small and larger issues) is below.

We thank the referee for their attention to detail and extensive feedback. We believe that the changes we are making in light of their feedback are indeed substantially improving the paper.

Language throughout the paper:

I find articles missing in a variety of sentences, maybe this is modern parlance but the language often comes across too casually and 'jargony'. I have noted a few concrete cases below.

We thank the referee for their attention to language detail and are copy-editing with careful attention to missing articles and casual language.

Notation: Pauli Z (or X, Y) is sometimes italic, sometimes roman, please be consistent.

We appreciate the referee's attention to formatting detail and agree it is better to be consistent, so we are updating these to be italic.

In a variety of places you write "logical error" as a placeholder for "logical error rate", for example page 1 "logical errors should be reduced ..", "demonstrating reduced logical error..".

Unfortunately, the phrase "logical error rate" is used inconsistently in the literature, so we are avoiding it except where the ambiguity is unimportant. In other works, "logical error rate" sometimes refers to what we call "logical error probability" (as in Fig. 3a), and sometimes refers to what we call "logical error per cycle." Both of these are "rates" from a certain point of view. We believe adding "rate" to the paragraph mentioned would not improve clarity, but we defer to the editor.

*In the Suppl. Material there are sections where references to Figs (say Fig. S9) are just written as "found in S9", better to consistently use Fig. S9 etc. This occurs at various places.
In several places in main text and suppl., you leave no space between word and a reference, like[1]. One occurrence of ref -> Ref.*

We have fixed these mistakes.

You refer to 'component errors' or 'component Pauli errors'. Even though I think I understand what you mean, it is not standard language, so being explicit about what you mean helps. For example, is cross-talk a component error? A circuit consists of elementary components (the planned gates and operations or idle locations) which all can fail/act differently than intended so I assume that these are component errors. In the caption of Fig. 4 you refer to 'operation errors' which I assume is the same as component error. The short hand "idle" or "idles" seem rather informal (meaning the circuit location during which a data qubit is idling).

We thank the reviewer for flagging this for clarification. Indeed, we are referring to the elementary operations within a circuit such as gates as “components.” We are adding a clarifying phrase to the caption of Fig. 1c. The referee is also correct to flag the inconsistent terminology “operation errors” in Fig. 4a, which we are correcting.

The question “is cross-talk a component error” is a subtle one. The way we’re phrasing it, all error associated with running a CZ gate is part of its “component error,” including errors resulting from stray interactions or control crosstalk. From the experimental point of view, the randomized benchmarking and cross entropy benchmarking experiments summarized in Fig. 1c are used to measure the “total component error”, which would include stray interactions and control crosstalk.

When we refer to “component Pauli errors,” we mean “component errors that are strictly Pauli errors on the qubits directly involved,” as opposed to other errors that could be associated with an operation such as leakage. We are rephrasing this sentence to help clarify.

We acknowledge the referee’s point about our usage of “idle” as a noun, for example “...and dynamical decoupling gates during qubit idles,” but we do not see an obvious improvement and defer to the editor.

Other typos/sloppy text in main text and Suppl Material:

Various occurrences of “the the”

Linbladian, Gottesmann -> Gottesman.

Typo “hypothesise.”

“A higher order effect cause by..”

“per gate and total of n..” -> “per gate and a total of n..”

“that Twirling Approximation...strongly over predicted..” -> that a twirling approximation ..strongly overpredicted...

“of the modeled error channels, (such as dephasing)”

Caption Fig. S20 Pauli+

“To estimate probabilities...” -> “To estimate the probabilities..”

“logical performance of our surface code qubits” -> “logical performance of our surface code qubit”

“Fitting error versus rounds to..” -> “Fitting error rates to extract the logical..” “developed data analysis..” -> “developed the data analysis..”

We appreciate the referee’s attention to detail and taking the time to point out these mistakes. Note we understand “hypothesise” is the British English spelling but defer to the editor. We use the plural “our surface code qubits” to refer to the five distinct logical qubits we test. Note we avoid the phrase “error rates” when ambiguous, such as this case.

Abstract

sentences are awkward and a bit sloppy. Here are some comments:

logical qubits within physical qubits” -> “encoding logical qubits into physical qubits”

..we report the measurement of logical qubit performance scaling.." -> "we report on logical qubit performance across.."

"we find our.." -> "we find that.."

"per round floor set...": although it is clear what is meant, is this 'floor set' a grammatical construction?

"begins to improve...illuminating the path..": what is begin to improve, did it improve or not? I am not sure in what sense the experiment is 'illuminating a path'

We appreciate the referee's diligent editing to attempt to improve these sentences. We respectfully disagree that the sentences are sloppy and defer to the editor on word choice within the summary paragraph.

Regarding the phrase " 1.7×10^{-6} logical error per round floor set by a single high-energy event:" to elaborate on our intent, we are referring to a floor (like "noise floor") in the quantity "logical error per round." We would be happy to rephrase this to improve clarity, but we do not see an obvious solution.

Regarding "begin to improve," we are simply phrasing this sentence gently to acknowledge that the improvement is small. This phrase also fits well with Fig. 3c.

Regarding "illuminating a path," the specific path is discussed in the conclusion in greater detail: *"Based on the error budget and simulations in Fig. 4, we estimate that component performance must improve by at least 20% to move below threshold, and significantly improve beyond that to achieve practical scaling. However, these projections rely on simplified models and must be validated experimentally, testing larger code sizes with longer durations to eventually realise the desired logical performance. This work demonstrates the first step in that process, suppressing logical errors by scaling a quantum error-correcting code – the foundation of a fault-tolerant quantum computer."*

Introduction

..suppress the operational..physical qubit overhead" -> "physical qubit and temporal overhead". It is known that the surface code leads to a slow-down of computation by a factor d where d is the distance, so this may be good to make clear. On that note, not even the logical error rate per round ϵ is the right performance metric demonstrating that one code is better than another, although we expect that this logical rate per round will decrease exponentially with code distance d , once one is below threshold. Demonstrating such exponential scaling (as you do for the repetition code) is of course the next milestone.

We thank the referee for pointing out the temporal overhead. We have rephrased this sentence in the introduction to highlight it. We agree there are various performance metrics to consider, and in this case we choose to focus on the logical error probabilities and logical error per cycle, as in Fig. 3. We agree that in future work, demonstrating this exponential scaling with larger, better-performing surface codes will be an exciting development.

"which traverse the lattice" -> "which traverses the lattice"

We have fixed this.

Fig. 1 and elsewhere, I am surprised that you do Hadamard gates and no $R_y(\pi/2)$ (and $R_y(-\pi/2)$)

gates or should I read Hadamard as $R_Y(\pi/2)$?

We physically implement Hadamard gates with a $\pi/2$ pulse followed by a virtual Z rotation. This has the benefit of simpler, more accessible notation and less bookkeeping for the reader.

Fig. 1c has no vertical axis label (!), no scale and just says "Cumulative distributions of errors", I assume it is fidelity or infidelity here. At the bottom of the caption, would it not be useful to refer to some of the Suppl. Material Section XI where the benchmarking is described in detail?

The quantities plotted in Fig. 1c are cumulative distributions. The vertical axis label would be "Cumulative distributions," but we moved it to the top of the plot for layout reasons. We find this is sufficiently clear, although we defer to the editor.

These are also commonly known as "empirical cumulative distribution functions" or "integrated histograms." Another way to phrase the vertical axis is "cumulative probability," ranging from 0 (none of the distribution is below this value) to 1 (all of the distribution is below this value). The horizontal axis, as labeled, is error probabilities (related to what the referee refers to as infidelity).

We are adding a reference to the supplement as suggested.

Caption Fig. 1 on measurement time: it would be good to list (average) time duration of measurement, single-qubit and two-qubit gates in a Table for clarity.

To clarify, we use the same durations for all qubits. We are adding the single-qubit and two-qubit gate durations (25 ns and 34 ns, respectively) for clarity. We think it suffices to write these briefly inline rather than make a table.

"nearest neighbours except on.." -> nearest neighbours except at.."

We prefer the existing phrase but defer to the editor.

T_2 , CPMG is measured at operation point or at a flux sweet-spot?

It is measured at the operation point, what we refer to as the "idle frequency."

*Paragraph at the end of Introduction on XZZX code:
"We also remove certain Hadamard..": don't you just move some Hadamard gates around and not remove?*

We do in fact remove Hadamard gates, not merely move them around. To explain this more clearly, we have added content to the supplement, including a new figure, S9. Briefly, consider the standard surface code with four CNOT layers (and some Hadamards). When we convert this circuit directly into CZ and Hadamard gates, there are many natural cancellations, most obviously between the second and third CZs, but also crossing cycles between the fourth and first CZs. As a result, every qubit does exactly two Hadamards during each (time-bulk) cycle. After these cancellations, the circuits can be reinterpreted as "measuring ZXXZ."

"bias in measurement error": you mean bias in readout error rate, i.e. measure qubit in 0 is more likely to be correctly identified as qubit does not decay during measurement. I don't see this discussed in Suppl. Material Section E, while it prompts you to choose a particular setup, so it is not unimportant.

We thank the referee for drawing attention to this important point. We are adding a remark to the supplement to indicate the bias quantitatively. Note this is a bit of a detail, but we wanted to include it as an “experimental best practice” to encourage adoption by the research community.

The XZZX may be less familiar to readers and has been proposed for dealing with biased noise, while here you just care about symmetrisation. One thing which is not clear is how it affects decoding. Usually MWPM on the regular surface code decodes X and Z errors separately. Are you still doing this now but 'reinterpreting an X error' as a Z error (and vice versa) when there was a Hadamard around the CZ gate in Fig. S12? I believe this is correct as you write in Section V in the main text about decomposing into disjoint error graphs (however I could not find this discussion in the Suppl Material on decoding). If so, I am also puzzled about the statement that Y errors cause more than 2 detection events in Section XIV. C in Suppl. Material. This is true for the complete error graph, but if we split the error graph into 2 parts and do matching separately, such Y errors correspond to proper edges.

This also relates to another issue about the description of the decoder: you refer to the “correlated MWPM method” in the Suppl. Material: “uses a variant of the two-pass correlation strategy detailed in [63]”. What correlations these are (Y errors being X and Z errors) and how correlations are used is never really discussed in Section XII. B. Please give some high-level idea about the correlated MWPM (after the BP algorithm).

This is a good question. The decoding error graphs are automatically generated from the circuits we ran on the hardware using an open-source tool called Stim (see <https://github.com/quantumlib/Stim>). Stim automatically produces what you call the “complete error graph” (or error hypergraph as we refer to it) represented as an object called a detector error model. Stim also performs an automated heuristic decomposition into disjoint graphs. For belief-matching, belief propagation is first run on this error hypergraph (passing messages between nodes and hyperedges) before it is split up according to this automated graph decomposition. Standard minimum-weight perfect matching is then run on these disjoint graphs. So, effectively, the reviewer is correct that certain X-errors are reinterpreted as Z-errors (and vice versa) with respect to the CSS surface code, although it is a small change that is handled automatically in our decoding pipeline.

We agree that the correlated minimum-weight perfect matching is not sufficiently described in the supplement, and have added a high-level description in the supplement (leaving a more full description to [63]). Essentially, correlated minimum-weight perfect matching works by first computing two disjoint minimum-weight perfect matchings, and then looking for single edge matches. These single edge matches often indicate a higher probability of a nearby error in the alternate error graph (e.g. a nearby Y-error that manifests as an error in the alternate error graph as well). After this first pass matching, the error graphs are reweighted using this information and minimum-weight perfect matching is rerun.

Figure 2:

Figure is quite dense, there is barely space for Figure h.

“...that manifest in detection pairs” -> “...that manifest in detection pairs, except at the boundary”.

The information in Fig 2a is hard to parse, in particular what are the pairs when they are timelike or spacelike separated (purple circles and blue circles).

We thank the referee for their close look and comments about Figure 2. We agree it is dense, but we feel the layout is appropriate.

The complete phrase is “*We show example errors that manifest in detection pairs;*” we are referring to the explicit examples shown, so we feel the remark about the boundary, which does not apply, is not needed.

We appreciate that this 3D spacetime information is difficult to communicate and took great effort to display it as clearly as we could. For example, the timelike pair is labeled in purple between two purple-highlighted bits separated by one time unit.

Caption Fig. 2: “...no preceding data idling”: is this really the reason and could it not be because of measurement errors. A measurement error causes defects in the current and the following cycle. At cycle $t=0$ measurement errors in that cycle can give a defect, but at $t=1$ there are measurement errors that happen at $t = 1$ as well as the defects coming from $t=0$.

The reviewer is correct that the lack of data idling error is not the only reason there are less detections in the first round. In particular, there are many possible error channels that would cause a defect in the following round (including measurement, as the reviewer points out). The only additional channel is the data initialization itself, which has a relatively low probability of error. We updated the caption to reflect this.

“The average detection...roughly 5 cycles [43]”: is this the number of cycles for the qubits to reach their leakage steady state? Could be discussed in more detail.

We agree this is an important point, and one easy to overlook, as the rise in detections is a weakness in the experiment. We attempt to be forthright about this by discussing it for a full paragraph. We agree it could be discussed in more detail but feel the current level of detail is appropriate given space constraints.

Specifically, the “characteristic risetime” is a $1/e$ risetime, so we would expect to take a few times that “to reach their leakage steady state.” Leakage mitigation techniques within the surface code are a subject of ongoing research.

For leakage removal the authors cite [42,47-49], but [42] and [49] only considers removing measure qubit leakage while from the Suppl. Material it is clear that data qubit leakage is important. Other work such as [78] proposes data qubit leakage mitigation, while <https://doi.org/10.1038/s41534-020-00330-w> studies aspects of leakage and leakage detection more generally.

It’s true that [42] and [49] are primarily concerned with measure qubit leakage, although in those results the measure qubit reset is also observed to reduce data qubit leakage, and we definitely need to cite [42] to provide some foundation for the repetition code experiment in Fig. 4b. Moreover, references [47] and [48] do consider data qubit leakage directly.

From our understanding, the additional reference the referee mentions (Varbanov et al.) is primarily concerned with deducing whether leakage occurred and potentially using this leakage detection to do leakage reduction on demand, which is an interesting idea. We have added this reference to the supplement alongside [78]. However, given bibliography constraints, we have not added it to the main text.

On Fig. 2h and matching with numerics.

First, the “ p_{ij} method” is not widely known and you should define the “correlation p_{ij} ” somewhere in the main text (and in the Suppl. Material Section which discusses this) as you do in [42]. More importantly, it is not wholly clear why this correlation p_{ij} is a relevant quantity: this is because these are approximately the error probabilities which lead to pairs of defects. Also this method was first executed in Spitz et al. <https://onlinelibrary.wiley.com/doi/10.1002/qute.201800012> (which is not cited). What are you doing with edges which only lead to one defect at the boundary. You are not plotting these probabilities anywhere or mentioning them, is your simulation also estimating these according to what you extract from the experimental data? Also, in caption of Fig. 2h: you mention diagonal edges here, but these are called spacetime like edges (ST) elsewhere and there is a class of unexpected spacetime like edges ST’. It would be better to stick with 1 naming convention here.

We agree, and have added the definition of p_{ij} correlations to the appendix section that discusses them. We thank the reviewer for informing us of the Spitz et al. work, and have added that reference. We have also added a formula for the edges at the boundary. The boundary edge probabilities are displayed at boundary tiles of the spacelike edge heatmaps, shown at the upper panels of Figs. S25 and S26 (look at qubit 1_5 for example). In the lower panels of these figures, we also show that Pauli+ simulations predict the boundary edge probabilities well. We also agree that our naming convention should be consistent; we updated the text to prefer “spacetime-like” uniformly.

On your claim that you match performance with simulation: in the main text in Section IV, the description of Pauli, and Pauli+ is quite brief and hard to understand what is really done, e.g. what does it mean to include ‘coherence information’?

We agree with the referee that this description is quite brief and can cause some confusion. With space limitations, we are not able to add very much, but we have rephrased this sentence, especially to help with the confusion about “coherence information.” To elaborate, we were referring to information about the coherence times of the qubits (T_1 and $T_{2,\text{CPMG}}$), which can inform whether the error has some bias beyond simple depolarizing error.

It is also worthwhile to dwell a bit more in the Suppl. Material on the disadvantages/weaknesses of your error model. For example, you add a depolarizing channel in order to match the observed error rate in RB/XEB that have no physical explanation (in Section XIV.A.5.1 Depolarizing errors).

We have added a discussion at the end of Section XIV.A.5 on the possible mechanisms that had to be accounted for by the depolarizing model, as well as some of the difficulty in translating the observed fidelities into errors.

Also, it seems that in simulation the leakage rates were uniformly picked while some qubits are clearly leaking more than others.

The referee is observant to point out that some qubits are leaking more than others, for example as visible in Fig. 2b. As we describe, the simulation reported here uses uniform leakage parameters. It would be more complicated but surely more accurate to use separate measurements on each qubit in order to refine the model, as the referee suggests. This is a subject of ongoing research.

Your crosstalk model used in simulation does not seem to be discussed (and you refer to later work)

The specific details of this model are beyond the scope of this work and our team is still working on refining our model, so we prefer to defer the discussion to an upcoming work. That said, we have added a few more details to the discussion at the end of XIV.A.4:

“Details of these simulations are deferred to a later work, though we note that the dominant error mechanisms stem from $|01\rangle \leftrightarrow |10\rangle$ transitions between neighboring qubits (undergoing separate CZ / idle gates). These transitions are highly sensitive to unwanted resonances arising during parallel operation.”

Another point to bring up here is that your Pauli+ simulation, although clearly an interesting and useful idea to better model the experimental data, cannot capture the fact of course that leakage is state-dependent (i.e. in a CZ $|11\rangle$ can leak to $|02\rangle$, but not the other states, and heating is expect to lead from $|1\rangle$ to $|2\rangle$ less indirectly from $|0\rangle$ to $|2\rangle$). You model this by assuming the qubit is in a random state and reducing the leakage rate correspondingly, but it is a feature that one cannot model easily.

The referee is correct that the Pauli+ simulation has several limitations, among them missing these state-dependent processes. Those limitations come hand in hand with the simplified computation, which made the large volume of simulations we conduct at the distance-5 scale tractable. We do mention another simulation technique (termed kraus_sim) in the supplement, based on simulating quantum trajectories. This quantum simulation explicitly includes the state-dependent nature of leakage, yet we find that, for our current experimental parameters, these details do not significantly impact the observed logical error rate or statistics of typical detection event edges. One justification for using just the “average” leakage rates is the fact that the data qubits are usually in a high-entangled stabilizer state, so that local error processes are effectively averaged over.

Section IV

“Measurement and reset errors...timelike pair”: also errors on measure qubits during the cycle can lead to such timelike pair. Similarly, for the spacelike pair there are contris from single and two-qubit errors. If you want to focus on the leading errors in this text, or just naming some example, please make this clear.

The reviewer is correct that here we are just giving examples of different possible error contributions, as mentioned in the prior sentence, but we defer to the editor on this point.

Section IV

You write “To estimate probabilities of detection event pairs..an appropriately normalized correlation p_{ij} ..” and you refer to [42], but you should also cite the Spitz et al. paper (see comment above). Furthermore, the sentence is completely unclear, what is appropriately normalized, you don't estimate the probability for a detection pair, you estimate the probability for a single error which leads to the detection pair.

As above, we now cite the Spitz et al. paper here. We will also refer to the supplement, which as above, now specifies the p_{ij} expression. We do treat these correlations as probabilities for subsets of detection events (using an ansatz probability model that is factorized according to a standard circuit-level error model), so each correlation includes many different individual errors (more precisely, the p_{ij} probability corresponding to an edge i - j is approximately the sum of probabilities of all clusters of detection events that contain the edge i - j). Probabilities for individual component errors are estimated from randomized benchmarking techniques.

Section V

Sentence “and each hyperedge is assigned its corresponding error mechanism probability” sounds very complex (i.e. “error mechanism probability”). The important point of this model is that it is a stochastic model and cannot capture coherent errors (amplitude versus probabilities) and that it assumes that errors at circuit locations occur independently, and this should be added.

For each hyperedge you estimate an error probability. I am not sure that the issue of hyperedges “generalized p_{ij} ” is described in a satisfactory way in the paper, the brief section Suppl. Section XVI. C is devoted to it and only sketches in a perfunctory manner how this works. The way it influences the results in the paper seems to be that it gives a better estimate for p_{ij} (avoiding negative probs) and thus the weights $-\log(p_{ij})$ in the decoder, like in [50]. I note that how the weights depend on p_{ij} is not even mentioned in Section XII on decoding. Trying to understand anything from Suppl. Section XVI is frustrating and does not help the quality of the paper so I would suggest that either the authors expand on describing this method well (if it is important in optimizing the decoder which is not clear from the text), or defer a solid description to another paper.

You write that the tensor network decoder depends on the “hypergraph error priors” (section V main text): are these ‘priors’ what comes out of the hyperedge analysis and how do they influence the tensor network decoder?

Overall, it seems that Section XVI should be better integrated to Section XII on decoding where priors are also mentioned (or refer in Section XVI for details covered by Section XII) and the “extended p_{ij} ” method is also discussed.

The terminology has precedence (e.g. ‘error mechanism’ described in the aforementioned ‘detector error model’ in ref [62]), so we’ve elected to keep it as is for consistency.

We appreciate the referee’s comment that generalized p_{ij} is insufficiently described in the supplement and have expanded on the description there. The intent of generalized p_{ij} is to provide probabilities for clusters of >2 detection events. This informs the probabilities of the error hypergraph – rather than the probabilities of the decomposed error graph – which is used in the BP stage of belief-matching, the tensor network decoder, and the rewrite rules for the two-pass correlated MWPM decoder. The referee is correct that we don’t mention how probabilities are converted to edge weights for the MWPM step, and we have added that to the supplemental decoding section. For the tensor network, the reviewer is correct that these priors are what come out of the generalized p_{ij} hyperedge analysis. The tensor network computes the probability of the logical error coset using the hyperedge prior probabilities.

Section V

“These graphs are decoded efficiently...minimum weight perfect matching”. From the Suppl. it is clear that the actual procedure is more involved, “correlated MWPM with some BP before this”, so this does not capture what is actually done. Without giving all details, the idea could be better presented.

Additional context: “The belief-matching decoder first runs belief propagation on

the error hypergraph to update hyperedge error probabilities based on nearby detection events [51, 52]. The updated error hypergraph is then decomposed into a pair of disjoint error graphs, one each for X and Z errors [34]. These graphs are decoded efficiently using minimum-weight perfect matching [53] to select a single probable set of errors.”

Here we are attempting to explain that we first run belief propagation to update the hypergraph and then use MWPM on the decomposed X and Z graphs. We believe this addresses the concern about explaining the procedure, “correlated MWPM with some BP before this.” We note that the second step of the decoder runs standard MWPM with the posterior edge weights generated by BP (rather than correlated MWPM).

Section V

You define the logical fidelity $F=1-2p_L$. For large numbers of cycles, one expects that the logical error probability $p_L=1/2$, essentially randomizing the logical qubit (see Fig. S14), and then the fidelity $F=0$, while a randomized qubit has fidelity $1/2$ with any pure state. So why not use $F=1-p_L$ which is the proper fidelity expression, assuming a logical depolarizing channel which maps $\rho \rightarrow (1-2p_L)\rho + p_L I$.

This ties in with the modelling in Suppl. Section XIII (see other comments below on that section), where you derive Eq. (2) and Eq.(2) does not follow using $F=1-p_L$ (since you go to fidelity=1/2 instead of an exponential decay to 0). Of course you are free to define “a fidelity” and just fit it with an exponentially decaying curve which goes to 0, but terminology may be confusing.

The quantity $1 - 2p_L$ is a good quantity to study. As a practical matter, it nominally starts at 1 and decays exponentially to 0, and on a semilog plot (such as Fig. 3b) this yields a line. We think this is the best way to plot this kind of data and hope the community will adopt it after seeing the nice lines in Fig. 3b. We also use this for our analysis: we actually fit a line to those points ($\log(1 - 2p_L)$ vs. t), an easy procedure with straightforward uncertainty analysis. Additionally, $(1 - 2p)$ comes up as a natural quantity in binomial probability calculations (the probability that a binomial random variable is even, related to our logical success probability).

We acknowledge the referee’s valid concern about someone expecting “fidelity to decay to 1/2,” which is thinking of “fidelity” for comparing two quantum states, though the situation here with the logical states and values is somewhat subtle. We clearly define it in both the caption and the axis label, so we do not feel it is confusing. If there is a better name for $1 - 2p_L$, we would be happy to consider it.

epsilon should be defined in main text instead of in Eq. (2) in this Suppl. Section, since it is the important parameter.

This is defined in the caption of Fig. 3c (“logical error per round ϵ_d ”) and in the main text (“We obtain a logical error per cycle $\epsilon_5 = \dots$ for the distance-5 code”). It is also redefined for clarity in Fig. 4c.

Section V High-energy event discussion:

You don’t post-select on leakage or high-energy events, but it may still be interesting to know what the error rates would be if you did so.

Since the high-energy events affect roughly 10^{-3} of the data and the measured logical error probabilities p_L are typically of order 0.1, we do not expect to see an effect from post-selecting

high-energy events from the surface code data. Please see below for more discussion about this post-selection in the context of the repetition code data.

Regarding leakage post-selection, while we acknowledge this may be interesting as a way to compare to other recent papers, there is a scalability problem as it pertains to analyzing our data. Suppose each measure qubit observes leakage in 1% of measurements. If we were throwing away instances where leakage was detected, for our distance-5 experiments, after 25 rounds we would expect only $0.99^{24 \text{ qubits} \times 25 \text{ cycles}} \approx 0.2\%$ of the data to remain.

Related to this you write "These events may be identified by spikes in detection event counts" but you don't specify in the Suppl. over what time window of runs you average the detection probability and throw out these runs when they are above some threshold, i.e you don't specify the procedure for tossing out those data.

We thank the reviewer for pointing out this oversight, and we are adding content to the supplement. To elaborate, a 50 cycle repetition code is executed in $40 \mu\text{s}$ and there is a delay between shots of $8 \mu\text{s}$, giving a total acquisition time of 24.5 seconds for the 500,000 shots. We apply a moving average filter with a window length of 30, identify peaks with DEFs above 0.2, and throw out 100 shots before and 500 after. In total, we discarded 741 shots (36 ms).

Concerning high-energy event issues, it is also not fully clear to me that this is an issue that occurs for other superconducting transmon teams (say the IBM group who has many processors in the cloud in continuous operation, so should know about this if present), or whether it can be mitigated by better shielding.

It would be difficult for us to comment on other groups' findings beyond what they have published or presented. That said, we can offer some remarks on how it might be possible that other groups with large superconducting devices have not reported this effect. We list several references below.

Firstly, this is a relatively rare effect, and is hard to discover without the appropriate experiment and data analysis. Standard experiments, such as the normal operating mode of IBM's public offerings, typically average many runs of an experiment together, and not necessarily adjacent in time. In [1], the authors studied the effects of radiation on qubits specifically, and did not report widespread quasiparticle poisoning events because their experiments were similarly averages over long time scales. When averaging, the overall impact of radiation is very small, with [1] reporting around a $1 \text{ ms } T_1$ limit, which is clearly non-dominant for even the best reported transmon qubits. It is only once you have time-resolved data that you can find the rare and short-lived burst events individually that it is possible to identify this effect at all. Indeed, at the most recent APS march meeting, research from IBM was presented where such a time-resolved experiment was run, and highly-correlated charge noise was discovered and attributed to radiation impact events [2].

Second, we see from published experiments that how the effect works is strongly dependent on device details. In our device, the qubits relax back to baseline performance on a timescale of around 20 ms. Contrastingly, the decay time of a smaller device with better thermalization in [3] was close to $200 \mu\text{s}$. Details such as the transmission of energy through bump bonds, bond wires, diffusion through the ground plane, variations in the superconducting gap and presence of other materials in the device stack can make a large difference to the theoretical

propagation of energy during an impact. It's not necessarily surprising that different groups with very different device architectures and materials have a very different or indeed much smaller signal to look for.

Regarding the question "can [impact events] be mitigated by better shielding", we believe the answer in short is no. Discussion of radiation physics is beyond the scope of this paper, but we are informed by other literature: In [4] the authors found that moving a superconducting resonator device to a low-radiation lab (Gran Sasso), including lead shielding improved the rate of qp burst events by around a factor of 10x.

[REDACTED]

This level of suppression is not sufficient for current plans to scale quantum computers to a practically useful level using QEC. Given the current error scale of an impact event is essentially unsurvivable, the event rate needs to be at least on the order of the computation time. Reasonable estimates for fault tolerant computations are generally measured in hours (c.f. [5] which proposes 8 hours for Shor's algorithm), so current event rates should need to improve by around 3000x from current event rates. Additionally, if chip areas grow approximately proportional to number of qubits, given that the event rate is directly proportional to chip area, the event rate will also increase by around 100,000x from this effect, which will need to be overcome as well. The 10x reduction offered by the use of lead shielding and a low-radiation laboratory is nowhere near sufficient to solve this problem at scale. It is also worth noting that full scale chip designs introduce additional radiation sources not accounted for in this experiment. While lead shielding can improve the background rate of gamma rays emanating from outside the fridge, sources of radiation in the packaging and the device itself cannot be shielded. With our current use of natural Indium (a beta emitter), it will be impossible to do better than the timescale of an hour or so, regardless of shielding.

Overall, even aggressive combinations of shielding and replacing fridge, wiring and device materials with radiopure versions is unlikely to achieve a reduction in error rate necessary for FT algorithms. Meanwhile, well-understood mitigations on chip (inspired by work on superconducting detectors, and now being studied and reported in qubits) should allow QEC to continue without any reduction in event rate, which should be more practical than shielding/radio-purifying.

Response references:

[1] Vepsäläinen, et al (2020) Impact of ionizing radiation on superconducting qubit coherence, <https://arxiv.org/abs/2001.09190>

[2] Thorbeck et al. (2020) Experimental characterization of correlated qubit decays in IBM Quantum processors attributable to cosmic rays and background gamma radiation, <https://meetings.aps.org/Meeting/MAR22/Session/M41.1>

[3] Wilen et al. (2020) Correlated Charge Noise and Relaxation Errors in Superconducting

Qubits, <https://arxiv.org/abs/2012.06029>

[4] L. Cardani, et al (2020) Reducing the impact of radioactivity on quantum circuits in a deep-underground facility, <https://arxiv.org/abs/2005.02286>

[5] Gidney, Eker (2019) How to factor 2048 bit RSA integers in 8 hours using 20 million noisy qubits, <https://arxiv.org/abs/1905.09749>

Section VI

On Fig 4: how can the fact that this only affects the logical error rate of the repetition code at very high distance be understood? I presume that the smaller distance codes are obtained from taking subsamples from the larger repetition code? But then a high-energy event, affecting say all T_1 , also affects the smaller codes, increasing the logical error rates there as well, so how do you understand this?

Perhaps this indirectly relates to understanding the curves in Fig.4d (for the surface code), namely in the cross-over region, the logical error rate seems to first go down and then sweep up again (see next comment).

The referee is correct that the smaller distance repetition code datapoints are taken via subsampling: all of the data points in Fig. 4b are based on the same 49-qubit, 50-cycle, 500,000-repetition dataset.

The referee is curious about why the smaller codes do not seem affected by the high-energy event (the light and dark blue points only separate around $d > 15$). Their mental model of the high energy event resulting in very poor T_1 is suitable. There is a brief period of time where all the codes perform very poorly and experience many logical errors. The largest code (distance-25) experiences almost all of its errors during that period of time, so cutting out the high-energy event results in a much lower logical error per cycle. The smaller codes experience logical errors much more often, around 10^{-2} to 10^{-4} per cycle, so cutting out the high-energy event doesn't make a difference.

We added a new plot to the supplementary information that we think sheds some more light on this situation and may be interesting to the referee, copied below. Please see the revised supplement for more information. Briefly, we can see a spike in mean detection probability (black dashed line), which is how we identify the event. At the same time, there is a spike in logical error probability (each experiment is 50 cycles long). All the distances spike up to a logical error probability of 0.5, and the larger distances recover faster. Note this is zoomed in around the event, a small slice of the whole dataset.

Figure 4d:

It is not clear that $s=0.9$ really is below threshold - is that clearly seen from the $d=25$ data point- it seems to be going flat, but it could also be a matter of scale. I think it'd be better to show.

The referee is correct to point out that “being below threshold” is a subtle subject, as any experiment or simulation will only be a finite size, in this case up to distance-25, so it’s difficult to say what might happen at even larger size. It’s possible the $s = 0.9$ curve would turn around at large enough size, since we are in the near-vicinity of the threshold, but we believe it’s below. Our purpose here is to provide more context for the meaning of “threshold” to avoid misunderstandings about our central claims.

More importantly, why would the logical error rate per cycle in the ‘cross-over regime’ first go down and then go up again as the curve seems to show? In the text you refer to finite size effects, but you don’t analyze it properly. It is useful to write an expected expression for ϵ_d and see how this matches your data. One would expect that ϵ_d should grow exponentially in d above threshold while decreasing exponentially in d below threshold, while it can have some prefactor, some function in d , which gives finite size effects. If this prefactor is monotonically increasing in d , I don’t think one can get the scaling you have found (i.e. dip and then increase). This is a bit puzzling since one expects that the prefactor consists of some combinatorial effect (scaling as $\text{poly}(d)$) divided by d since ϵ_d is the error rate per cycle. One reason that there is first a dip in logical error rate and then an increase as a function of d is that the decoder for large d is not decoding up to the distance really (because computational cost will surely increase with d). This would be an important problem. Hence I believe that the authors should work on understanding/explaining this data in a more convincing manner (or put error bars on the curves if the shapes of the curves are misleading).

We appreciate the referees concern here, but we believe the simulation and decoder are correct. These finite-size effects have been observed in several works, e.g. in <https://arxiv.org/abs/0905.0531>, <https://arxiv.org/abs/1202.5602>, and <https://arxiv.org/abs/1311.5003>. As for the simulation itself, error bars are well within the points, as each point was obtained from Monte Carlo sampling on the order of $\sim 10^9$ rounds per point for accuracy.

A closed form expression for the precise number of possible logical errors at a given code distance is not expected to exist; instead, one uses stochastic simulations. Taking a look at the threshold region in Figure 10 of another work <https://arxiv.org/abs/1202.5602> (<https://doi.org/10.1103/PhysRevA.86.042313>), one can see that the rightmost distance-55 data point lies above distance-3, while the distance-11 data point lies below distance-3. This demonstrates the characteristic behavior described above, and the next two lower distance-55 data points exhibit similar behavior, namely initial error suppression that doesn’t last.

The correctness of the decoder can be verified as minimum-weight perfect matching itself has mathematical checks that can be used to confirm a minimum-weight matching has been obtained, and such checks have been performed exhaustively on stochastic data for many CPU years.

To understand why this turnaround happens, we turn to the analytic proof of a threshold with matching in <https://arxiv.org/abs/1206.0800> (<https://doi.org/10.1103/PhysRevLett.109.180502>). Referring to the discussion before equation 3, one can upper bound the number of paths along which logical errors can occur. When the code is small, most nodes can only be exited in a few

directions due to the presence of nearby boundaries. This means that as the code distance grows, not only does the number of paths along which logical errors can occur grow exponentially, but the rate of exponential growth itself accelerates during the transition from small code distances to larger ones. A low probability of error that initially leads to an exponential suppression of the probability all logical errors can fail to suppress errors as the rate of growth of the number of possible logical errors accelerates. However, the acceleration itself is bounded, meaning a threshold error rate still exists, and exponential suppression will occur at a sufficiently low error rate. We added a reference to the proof of threshold for matching for those interested in these details.

Could you provide a definition of the error suppression factor Λ ? (you refer/introduce it in Section VII but it is hard to find). In Section V you give it with a subscript $\Lambda_{d/d+2}$ and you mention it in Section VII without subscript (so any asymptotic large d version of it?). In Section XI.G there is also a Λ_{35} typo and in that section it also mentions Λ without subscript.

Yes, we define $\Lambda_{d/(d+2)} = \epsilon_d / \epsilon_{d+2}$ in Section V, as the referee notes. Based on the referee's suggestion, we can promote the inline expression to an equation so it stands out more. We thank the referee for pointing out the inconsistent notation; we are replacing bare Λ with $\Lambda_{3/5}$ and $\Lambda_{d/(d+2)}$ as appropriate.

Section VII

You write "For efficiency, the simulation considers only Pauli errors" So why not Pauli+, is this not efficient enough?

That's correct: our Pauli+ simulator is not as fast as our Pauli simulator. Due to the size of the simulation (a 2D scan over distances and error rates with high accuracy), we elected to use the Pauli simulation.

Section VII

One of your main conclusion is that your experiment "lies in a cross-over region...". As I mentioned above, if this is one of your conclusion it needs to be substantiated with a better analysis of the finite-size effect behavior. This is also important for interpreting what is seen in future experiments.

Please see above for more discussion about the crossover regime and Fig. 4c-d. We believe the referee's concern here is just about the existence of this crossover regime. The central claim in this discussion is $\Lambda_{3/5} > 1$, an empirical claim. By definition, this lies in the crossover regime, *unless* it is below threshold, but we do not believe it is (we would view that as a much stronger claim).

You write "Based on the error budget...component performance must improve by at least 20%..": does this somehow obviously follow from the curves for differing s ? Could you clarify this? Also, is it just device performance or also decoder performance which needs to improve?

Full context: "Based on the error budget and simulations in Fig. 4, we estimate that component performance must improve by at least 20% to move below threshold, and significantly improve beyond that to achieve practical scaling. However, these projections rely on simplified models and must be validated experimentally...."

This is a rough estimate. The error budget in Fig. 4 refers to Fig. 4a, where improving each of those contributions by 20% would (in the linear approximation) reduce $1/\Lambda_{3/5}$ by 20% (increasing $\Lambda_{3/5}$ by 25% to around 1.3). Based on the simulations in Fig. 4c, we think that would put us just below threshold ($\Lambda_{d/(d+2)} > 1$ for all d). Another way to look at this is in Fig. 4d, where $s = 1.2$ is at $\Lambda_{3/5} = 1$ (slightly worse than our experimental results, but close), while $s = 1.0$ is very close to threshold ($\Lambda_{23/25} = 1$).

Here, we are referring to device performance. On the subject of decoding, improvements would help achieve large-scale quantum computation, but that is not what we reference with this 20% estimate.

General comment on information about temporal execution of the experiments:

I believe that it would benefit the community if more information is provided about timescales of fluctuations in parameters and over what period of time data has been taken. In Section XI.B you write about random fluctuations due to TLS, but no typical time-scales or information about how frequent one re-optimizes versus the overall time for data acquisition. In Section XI.G.3 you provide some details related to ordering of experiments, but over what period of time data are taken and how much re-calibration is done in between is not made clear.

We thank the referee for drawing attention to this important point. We are adding information about the duration of the surface code experiment data acquisition to the supplement in Section XI.G.3 (it's about 1.5 hours). Regarding the timescales of TLS fluctuations, we would direct the referee to Ref. [59] (Klimov et al., *Phys. Rev. Lett.* 2018) which studies this in great detail.

Regarding “how much re-calibration is done in between,” the only item is the occasional “frequency update,” as described by the pseudocode in Section XI.G.3. No other re-calibration occurs during the surface code experiment data acquisition.

Sect XI

Section XI.B. shouldn't f_10/21 be f_11/20 as that is the avoided crossing which the CZ uses. “distance-5 objective”: what do you take as objective here? The performance of parity checks?

Here, we are making the statement that the $0 \rightarrow 1$ transition of one qubit is being brought on resonance with the $1 \rightarrow 2$ transition of the other. We have clarified the notation, and also added some more information about the “surface-code objective” being optimized.

Section XI.C

Are you determining L1 of each single qubit gate and/or two-qubit gates as well? For single qubit gate you write a whole section XI.F on reducing microwave cross-talk which induces leakage, but you never give what you measure as L1. Similar question on L1 during readout (which contributes to readout error in Fig S9).

Do you measure other parameters like the leakage swap or the phases leaked qubits impart on computational qubits if those are measured (phases are mentioned in Section XIV.A.3). If these parameters are used in the error model in simulations, they should be provided.

We measure the leakage in both our single and two qubit gates. The median (mean) leakage per gate after compensating for microwave crosstalk for single qubit gates is around $2e-5$ ($5e-5$). The typical mean leakage during two qubit gates is around $1e-4$ per gate. We have added this information in sections for single and two qubit gates.

We do not explicitly measure the impact of unwanted swaps and controlled phases from neighboring qubits on data qubits.

Fig. S7

The fig refers to T_2 but you used T_2^{CPMG} before, so denote by the same symbol?

Quite a lot of qubits are breaking the $T_2=2T_1$ limit. I guess this can always happen if there are fluctuations and recalibrations between the times these datasets were taken, but I think it is useful if the authors comment on how they understand this.

We have updated the caption to specify $T_{2,\text{CPMG}}$. The reviewer is correct that cases where $T_2 > 2T_1$ can be attributed to fluctuations in both the measured T_1 and T_2 as described in the Optimizing gate parameters section. We have added a short explanation of this in the section on single qubit gates.

Fig S8 and S9

Please provide definition of coupling efficiency and measurement efficiency. Define readout-resonator coupling in % .

"CZ...as measured in the surface code layers" is not clear. Figures occur quite late after associated text, so move up.

You don't give reset error probability and describe what reset does: does reset include reducing leaked states to computational basis states?

We have clarified the discussion around CZ gates and readout-resonator coupling efficiency. The reset operation is described in other papers cited in the text; accurately measuring the reset error in the presence of SPAM is an ongoing area of research.

Section XI.E

"swapping between neighboring qubits", does this imply all transitions among the 4-level systems, i.e. 01-10 and 12-21 and 21-03 and 12-30, more or less?

We have clarified that this refers only to 10/01 and 11/02 swapping.

Section XI.F.

Define/introduce receiver and source qubit before using it. Are source and receiver qubits neighbors or one only needs to have f_{01} close to f_{21} .

"We apply a...for V'..." -> "We apply a...with V.."

To clarify the definitions of source and receiver, we have added the following sentence: "In such a process, a small amount of the microwave signal from a "source" qubit couples to a "receiver" qubit which can then experience a crosstalk error."

Section XI.G

"We present the surface.." -> "We have presented.."

"..which minimized the detection rate in parallel..": do you mean average detection rate (over time or over measure qubits?) or the detection rate/probability for each measure qubit. Be more specific.

We have expanded the discussion of the phase calibrations, and it should be clear now that we are optimizing on the detection probabilities of specific detectors for each phase correction.

Section XI.G choice of $d=3$ grids

“excess phase errors”: Excess compared to what and for what reason? Did T_2 go down by what?

“our models”: what simulation model are you referring to here?

“outlier qubits” -> boundary qubits

We have clarified that the phase error refers to a miscalibration of the aforementioned phase correction. The simulation model refers to Fig 4(c-d). That outlier qubits are in reference to a single under-performing qubit having a larger impact on a distance-3 code than a distance-5 code (“outlier” as in a distribution, not referring to its physical location).

Caption Fig. 11

You refer to ‘moments’ and ‘single moment’ (which may relate “Moments” in Cirq), but steps is more standard terminology.

We have elected to keep “moment,” but we defer to the editor.

“Note although..” -> “Note that although..”

We have fixed this.

Section XI.G or elsewhere:

Somewhere it can be mentioned how to get the data for $d=3$ from $d=5$, i.e. set the other qubits to 0 (or +) so that weight-4 parity checks only depend on the data qubits of the $d=3$ code. This can lead to extra error at the boundary if the non-participating qubit needs to be in +. Or are you doing a different experiment with weight-2 checks for the $d=3$ cases? Please clarify, perhaps in main text. This comment also applies to the repetition code.

Fig. 2e shows the configuration of the stabilizers for the distance-3 codes, which we measure in separate experiments (indeed implementing weight-2 stabilizers). Please see the pseudocode in section XI.G.3 for further clarification. We do not subsample the surface codes.

In contrast, data for smaller repetition code distances are subsampled from a single distance-25 dataset (see Fig. 4 caption).

Section XII Decoding

“To account for this...”: do you mean a Pauli-twirled version of T_1/T_2 , but what duration is picked, I presume the duration of the location which represents the edge. Nothing is mentioned about boundary edges here, i.e. error events which lead to a single detection event and how well and how these are estimated by this method. Typo “averaged over round”

For simplicity, we use a uniform lower bound determined by the average T_1/T_2 times to produce a depolarizing channel whose strength is given by the duration of the measurement time; we’ve added the duration. Using a more precise per qubit lower bound determined by individual twirled

T_1/T_2 channels may yield an improvement, although we expect it may be small (see discussion at the end of Section XII.A).

We have fixed the typo and added a parenthetical to note that boundary edges correspond to the ‘one-body’ correlations mentioned in the text. These are frequently the edges that must be reweighted due to negative probabilities. The reason is that the generalized p_{ij} method used for decoding – as has been added to the text – takes a ‘top-down’ approach: a k -body correlation is computed by subtracting the probabilities of all $>k$ -body correlations that contain it to avoid overestimating its probability. This means that the boundary edges – which are composed of a single detection event – are susceptible to ‘over-subtracting’ as they’re contained in very many higher order correlations. This can occur when the true probability distribution of detection events contains correlations that are not present in the original probability model ansatz used to compute the generalized p_{ij} (for example, due to leakage or some other ‘out-of-model’ error).

A toy example illustrating this is a perfectly correlated distribution on three bits, decomposed assuming an ansatz of all six 1- and 2-body correlations. Because there is a 3-body correlation present in the true distribution but not the ansatz, the 1-body terms will be assigned a negative probability as we would ‘over-subtract’ due to dependent 2-body correlations.

Section XII.C

“error hypergraph is decomposed into..”: more details would be useful. You write “Practically, this decoder...”: the statement does not refer to what size code you are decoding, so I assume you believe it works for any d ? (but this would require parallel execution). If not specify what d .

The decomposition is computed using heuristics within Stim - we will add a reference. The referee is correct that we mean this statement to apply to any distance d using parallel execution, which we believe doable due to the decoder’s ‘local’ flavor (BP plus a MWPM engine similar to that described in <https://arxiv.org/abs/1110.5133>).

Section XIII

Please say what r is (number of rounds).

Please write what N is

You could provide some references where modeling is done differently. The paragraphs on quantification of out of model errors are interesting but hard to follow, it just reads as a short hand description of some numerical procedures which a reader cannot verify nor follow for correctness really. Not sure how easy it is to improve this.

We have clarified the variable names. N is the number of samples. In terms of other references, e.g., [22] fit an exponential decay of logical X and Z to extract a logical T_1 and T_2 , which is equivalent to analyzing our X and Z basis experiments separately. As far as we can tell, they don’t give many details on the fit (that is, whether they do a least-squares fit with uniform weights of the exponential decay or the log of the exponential decay, or with binomial reweighting as we do), or discuss the error bars on the logical error rate (that is, whether they propagate binomial errors through the fitting procedure, use excess residuals as we do, or do something different).

Section XIV

“depolarizing channels each device operation”. Experimentally characterized fidelities: could you be clear what data of what Section is used and refer to it.

We have fixed the noted typo. We added a reference to the values in Fig 1c of the main text and Table 1 in the supplemental: *“The typical fidelities observed are summarized in Fig 1c of the main text. Mean values for the $d=5$ experiment can be read off directly from the first rows of Table 1 (excluding leakage and crosstalk).”*

On brute force, quantum trajectory simulations: these are done in some other papers as well, in particular for modeling leakage and logical performance for the surface code, see e.g. [84].

At the end of Section XIV.B.1 we have added this citation and two subsequent works, though we note that those were based on a density matrix simulation and not pure state quantum trajectories.

Section XIV.A.1.

H_{idle} is not defined, just give an expression/description for it.

We have added a sentence below the relevant equation summarizing H_{idle} : *“where H_{idle} denotes the Hamiltonian of the idling transmon (diagonal in the standard basis $|j\rangle$).”*

Section XIV.A.3

“in characterization” -> “in the characterization”. What is meant by isolated CZ gates? Just CZ gates? Or CZ gates not run in parallel when other gates are run?

“The qubit ordering above matches the f_{10} transition frequencies...”: I don’t follow this sentence, how does qubit ordering in an equation influence what qubit leaks.

The mentioned typo has been fixed. To clarify “isolated”, we have added the following note: *“To exclude parallel gate crosstalk effects, leakage in each CZ gate was measured while all neighboring qubits were held idle.”*

We have replaced the qubit ordering sentence: *“Note that during implementation of the diabatic CZ gate, only the higher idle frequency qubit occupies the $|2\rangle$ state. Therefore this is the only qubit with considerable dephasing-induced leakage. The qubit ordering used to specify the transition Kraus operator K_1 reflects this”*

Section XIV.A.4

You mention 2 sources of cross-talk: inactive couplers and ‘diagonal capacitive couplings between next-nearest-neighbor qubits’. I don’t know what is ‘diagonal’ about capacitive couplings. Aren’t these all treated/leading to ZZ interactions?

In the sentence describing diagonal couplings, we have clarified that these qubits are *“separated by a single horizontal and vertical displacement on the square grid.”*

Additionally, at the end of the paragraph we have noted that *“the dominant error mechanisms stem from $|01\rangle \leftrightarrow |10\rangle$ transitions between neighboring qubits (undergoing separate CZ / idle gates). These transitions are highly sensitive to unwanted resonances arising during parallel operation.”*

Section XIV.A.5 “fidelity decay curves are converted to an inferred Pauli error”. How? You get an estimated Pauli error from physical modeling and then you have to add some extra depol noise to get the fidelity, but doing this is not unambiguous.

We have added the clarification: “...converted to an inferred Pauli error: we find the depolarizing channel that would produce the observed same decay curve and extract its Pauli error.”

We have also added the following statement about the ambiguity regarding fidelity: “We assume that for independent error sources, this should match the total Pauli error, to leading order.”

This statement is correct as long as each independent error source is approximated well with a linear contribution to the total Pauli error. It is an exact statement (true to all orders) if the error sources are modeled as Pauli channels: the Generalized Twirling Approximation reduced our quantum channels to Pauli channels, and in the supplemental we found good agreement between this approximation and the quantum simulation.

Section XIV.B. Simulator Details

Section XIV.B.2.

“A perfect rest to a single-qubit device $|0\rangle$ state”: in this section you choose to refer to the qudits as devices, this is ok as a choice but could be mentioned at the start.

We have added the following clarification at the end of the second paragraph:

“Note that throughout this section, we will refer to each multi-level transmon qubit as a ‘device.’ Additionally, we refer to the local energy eigenbasis of each transmon qubit as the ‘standard basis,’ with states $|0\rangle$ and $|1\rangle$ forming the span of the ‘computational subspace.’ In the context of multi-device operations, the computational subspace will refer to the tensor product of the relevant devices’ computational subspaces.”

In eq. (15) why don’t the Pauli operators act on qubits in D (they transition down and now get a random Pauli on them.)

In hindsight, a more accurate simulation would have indeed done this. We have added the following discussion at the end of equation (13):

“... but we note that a more physically accurate distribution could be used here. To account for the fact that transitions occur in the standard basis, we may instead randomly prepare the device in one of the σ_z eigenstates. In the context of Twirling Approximation (Sec. XIV.B.3), the probability of preparing one of the states could be set to match the device’s (σ_z) observable, on the condition that the device decayed to the computational subspace.”

Considering that the existing classical simulation results agree well with the quantum simulation, at this point we are hesitant to rerun all of the simulations using this relatively small change.

I find the pseudocode description not superclear. When you introduce \hat{G}_i (first with hats, then without.), include $G_{i[j]}$ as the j th entry of this Pauli being I, X, Y or Z .

We fixed the inconsistency with the carat symbols on the stabilizer generators and made adjustments to the language in the pseudocode to improve clarity. We have added the following sentence to clarify the j th entry: “(Note that $G_{i[j]}$ in $\{I, \sigma_x, \sigma_y, \sigma_z\}$ refers to the component of G_i acting on device j .)”

Additionally we reference the original tableau stabilizer implementation (Improved Simulation

of Stabilizer Circuits, Aaronson and Gottesman, 2004) where many of the processing steps are the same.

Section XIV.B.3

Good to be concrete in how you get the twirled channel, but some mention on how to extract Kraus operators from the Choi state which characterizes the channel could make this section be more embedded in standard QI theory (instead of reading like pseudocode)

We appreciate the referee's feedback, but feel that the discussion here assumes that Kraus operators are already given. Section XIV.A gives an explicit description of how the Kraus operators are computed for each error channel. Extraction of the Kraus operators from the Choi state is referenced at the end of section XIV.A.1.

We have added an additional sentence with explicit details: *"These can be extracted directly from any square root of the Choi matrix, $\sum_{ij}(|i\rangle\langle j|) \otimes E(|i\rangle\langle j|) = X X^\dagger$ "*

QMC simulation, does this stand for Quantum Monte Carlo? Isn't this just full density matrix or 'brute force/quantum trajectory simulation that you mention earlier, QMC suggests other things to me and the abbreviation is not even written out when it first occurs/is described in XIV.B.1.

We agree with the referee that mention of QMC can lead to confusion. We have replaced both mentions of "QMC" with "quantum trajectories." We have added an additional reference of the kraus_sim simulation as a "quantum trajectories" simulation to emphasize that this is the only kind of quantum simulation done in our studies (density matrix evolution is never carried out).

Fig S21

Would be interesting to see the same plots but for Pauli+ and experimental edges. I think this is particularly important since the Pauli+ are used for error budgeting and "physical understanding".

We agree with the referee that it is interesting to compare edge probabilities from experimental and Pauli+ data. We have done such a comparison in Section XVI.A-B.

Section XV

Here you refer to ϵ_L (L stands for logical now?) while elsewhere you use the subscript ϵ_3 and ϵ_5 .

We thank the referee for noticing the notation inconsistency. We have changed the notation from ϵ_L to ϵ in Section XV.

Why is the half-operation point a relevant point to evaluate sensitivities at?

When we construct the $1/\Lambda$ error budget, we assume that $1/\Lambda(p)$ can be well approximated by a quadratic function of the error vector p : $1/\Lambda(p) = p^*(a + H*p/2)$, where a is a vector and H is a matrix. At the experimental operation point p_{expt} , the $1/\Lambda$ value can be then written as p_{expt}^*w , where w is the weight vector $w=(a + H*p_{\text{expt}}/2)$ that in turn is the gradient of $1/\Lambda(p)$ at half-operation point, $p_{\text{expt}}/2$. This is the reason why we evaluate the sensitivities of $1/\Lambda$ at half-operation point (to compute the weights w) when evaluating the error budget of $1/\Lambda$.

Section XV.A paragraph "In particular, the SQ gate..." is a repeat of information mentioned elsewhere.

Although we agree with the referee, we prefer to mention this information again for clarity. We

have added a reference to section I, where experimental details are given. This paragraph now begins as: “As discussed in Section I, SQ gate errors are ...”

Abbreviation Dq in Figs S22/23 not previously used.

We thank the Referee for spotting this; we have replaced ‘Dq’ with ‘data qubit’ there.

Section XVI.A : perhaps you can point out where hyperedges due to Y errors occur in Fig. S24 (as you mention them earlier..)

This is a good clarification. We have added some annotations in Fig. S22 (upper panel) to indicate features in the p_{ij} data related to Y errors (see black dotted lines with the central data qubit, for example).

Section XVI.B time-like edges. You mention that leakage and CZ also contribute to T edges, but don’t say anything about the mechanism.

The contribution of CZ errors to T-edges can be understood by propagating Pauli errors through the QEC circuit. For example, consider a single Z error on a measure qubit during a CZ. CZ errors also contribute to S edges and ST edges.

We discuss plausible mechanisms of leakage in Section XIV.A (noise models) and briefly in Section XV.A. Leakage in the data qubits (which can come from CZs) produces correlated detection events at the neighboring measure qubits, which in turn leads to error edges that include the usual edges (e.g., T-edges) and unusual edges such as ST’ edges.

Section XVI.B space-like edges: clarify what an SX edge is.

We thank the referee for spotting this; we have explained the meaning of SX edges (spacelike edges between 2 X measure qubits) at the beginning of Section XVI.B.

*“and at some round t” -> “at some round t”
“also contributes..” -> “also contribute”
“T1 qubit decay” -> “T_1 qubit decay”*

We have fixed these typos; we thank the referee for noticing this.

Fig. S29 leakage in D qubits, what are D qubits?

We have replaced “D qubits” by “data qubits” in the title of Figs. S25, S26.

*XVII
“use error probability..” -> “use an error probability”*

We have fixed this typo; we thank the referee for noticing this.

You mention that most likely underestimate leakage in simulation (I presume you mean leakage of qubit 2_4 and 2_6): do you any idea that this is due to heating or due to CZ gates, or yet some other mechanism.

Yes, the referee is correct that we mean leakage accumulation in data qubits 2_4 and 2_6. We

believe it is caused by imperfections in the CZ gates acting on those data qubits.

“solving numerically this nonlinear systems..” -> “solving numerically this nonlinear system”

This text has been replaced.

Section XVII

Perhaps use vector notation in p_{ext} and w for clarity.

“use error probability..” -> “use an error probability”

We now use vector notation for p_{ext} and w , and have fixed the typo.

Section XVIII

The table is representative although I find your description of bosonic error correction (“embedding QEC into the physical qubits”) a bit misleading. Maybe better to say “embedding QEC into the definition of a physical qubit as a subspace of an oscillator”

We thank the referee for this clarification, which we have adopted.

Response to Referee #2

General comments

The authors of “Suppressing quantum errors by scaling a surface code logical qubit” realize a distance-5 surface code logical qubit and demonstrate that its logical error rate is slightly lower than the average of the logical error rates of four distance-3 logical qubits implemented on the same device. Accurate modeling of the experiments, including leakage errors, allows the authors to understand the main error sources limiting the logical performance. Furthermore, the authors run a distance-25 repetition code finding no obstacles for achieving algorithmically relevant error rates in the future except for the impact of high-energy radiation.

The authors emphasize the modest observed performance gain from $d=3$ to $d=5$ and underpin it with a thorough statistical analysis in supplementary information. In my opinion, the work would not lose in importance, if e.g. only equal logical performance was demonstrated. In fact, the number of qubits is almost tripled from $d=3$ to $d=5$ and so is the number of operations executed in parallel. Maintaining performance in such a setting is a major experimental achievement and raises hope that fault-tolerant quantum computers based on a surface code architecture can be built in the future. Prior to this work, only distance-3 surface codes had been experimentally realized using superconducting circuits (Refs. [22] and [24] of the paper). Another strength of the paper is the thorough modeling of the experiment, including not-so-obvious-to-model but important error sources such as leakage and crosstalk. Understanding them in detail and determining the main error contributions to the logical error rate measured is indeed very important to improve device performance in the future.

We thank the referee for their clear summary of our work and their appreciation for the technical challenges involved in scaling up to distance-5, along with the importance of this demonstration as a major experimental achievement.

The main body of the text is accessible and clearly written and the conclusions drawn appear to be correct. The supplementary material supports the claims of the paper and provides most of the details required for an in-depth understanding of the experiment and the simulations.

We thank the referee for their comments on the manuscript presentation and scientific validity.

However, providing more detail on the device architecture and its implementation would be essential for any other party to be able to reproduce the results presented in the manuscript.

We appreciate the point from the referee and are adding more information based on their detailed recommendations below.

As noted, we use a very similar device and experimental setup as in Refs. [42] (Chen et al., *Nature* 2021) and [32] (Arute et al., *Nature* 2019) but at a larger scale. The details provided in those references were sufficient for replication; see for example Wu et al., *Phys. Rev. Lett.* **127**, 180501 (2021), <https://doi.org/10.1103/PhysRevLett.127.180501>.

Our main criticism concerns the following points:

1. We find that the explanation and motivation for using the ZXXZ surface code is not sufficiently detailed. The authors motivate it with symmetrizing X- and Z-error rates, but it is not clear to us why this is desired. On the contrary, we would find it interesting to distinguish between the impact of data qubit energy relaxation errors and dephasing errors on the logical performance. The authors do not comment at all on the more

conceptual implications of using the ZXXZ surface code: Is it expected to perform better than the standard surface code on their device / have the authors run the standard surface code on their device? What are the differences regarding the decoder in the view of the material presented in Ref. [46] by Ataiides et al.? Also the logical operators of the ZXXZ surface code are not presented. We feel that a separate section in the supplementary information on the ZXXZ surface code would be very helpful. We understand that focusing the discussion on the standard surface code in the main text is more pedagogical, but we would recommend making connections to the actual code implemented whenever possible.

We appreciate the referee's attention to detail here. We agree that discussing the standard surface code in the main text is more pedagogical and accessible. Most of the literature on XZZX codes (e.g., Ref. [46], Ataiides et al.) is about exploiting extreme bias, which is not what we're doing, so this can be a point of confusion.

We would contend that the ZXXZ circuits we run are the natural result of compiling the standard surface code circuits with four CNOT layers (per cycle) into CZ and Hadamard gates. The primary difference is the cancellation of adjacent Hadamard gates from consecutive stabilizer measurement rounds, which can then be reinterpreted as "measuring ZXXZ." We have added a new supplementary figure (S9) to illustrate this clearly.

Each Hadamard is implemented as a $\pi/2$ microwave pulse and virtual Z rotation. It is important to get the phase of that $\pi/2$ pulse correct to avoid injecting errors. We optimize those phases in context (see discussion in Sec. XI.G.1). By removing unnecessary Hadamards, we reduce the number of opportunities to inject errors and simplify the phase optimization substantially, which we expect to improve performance. We have run experiments with various compilation strategies, but we have not optimized this in great detail, and this is a subject of future research.

It's true that there are also implications for how data qubit idle errors are detected (which measure qubits see X or Z idle errors). These errors represent about 20% of the budget in Fig. 4a, and the others aren't affected by this ZXXZ compilation.

For our experiment, we think symmetrizing the logical X and Z error rates is desirable. As a practical matter, our main objective here is to compare two code sizes (distance-3 and 5). Matters are already complicated enough with the inhomogeneities across the device and running experiments on various distance-3 codes to cover the distance-5 code's qubits. Any substantial difference between logical X and Z error rates would make the comparison even more difficult. Moreover, a real computation would be partly limited by the worse basis.

We had tried running the standard surface code without this optimization previously - it typically resulted in a mild asymmetry between the X- and Z- logical error rates, which was reduced after the switch to ZXXZ. No changes were made to the decoder pathway, as it was agnostic to the Hadamard cancellation.

Regarding logical operators, referring to Fig. 1a, Z_L would read (black circles, top to bottom) ZXZXZ, while X_L would read (green circles, left to right) XZXZX, referring to the data qubit state

between cycles. Based on the referee's recommendation, we are adding content to the supplement to describe the compilation and logical operators in greater detail.

2. Little details are provided for the repetition code experiment. The data set presented suffered from a single high-energy event: How representative is this data set? How long was the data acquisition time and what is the number of high-energy impact events expected during that time based on the authors' previous study? Generally, we think the repetition code does add only little to the already nice achievements of the distance scaling experiment. The general observations presented in the two paragraphs of this section are rather short and have been stated in related previous work by the same team published in the same journal. Also, the impact of high-energy disturbances is not discussed in much detail here, while it is being discussed by the same team in other papers. If needed for space constraints, the whole section could be moved to the supplementary material, for example.

The referee is correct that the repetition code experiment has a close relationship with previous published work from our group, and this is why we devote relatively little space to it. However, it is operating at a much larger scale (distance-25) and much lower error per cycle (10^{-6}) than any previous work. We find this interesting as a tool to probe for potential logical error "floors," such as the one presented by the high-energy event in this case. In a sense, we can preview what future higher-performance, larger-scale surface code data might look like, with the simpler repetition code experiment.

A 50-cycle rep code is executed in 40 μs , and there is a delay between shots of 8 μs , giving a total acquisition time of 24.5 seconds. Surveying 10 datasets from various repetition code experiments taken around the same time, we observed an average of 2.3 cosmic ray hits per experiment, indicating that a cosmic ray hits our processor once every 10.6 seconds. This is consistent with our previous work in Ref. [56] (McEwen et al., *Nature Physics* 2022). Note that there are differences in experimental setup, including the processor size.

3. The readout architecture of the device is not sufficiently documented (Purcell filters, multiplexing, use of near-quantum-limited amplifiers etc.). Furthermore, in the experimental details section of supplementary information, some standard parameters such as single-qubit and two-qubit gate durations are missing, and some of the parameters are stated in a non-standard way without defining them (effective qubit-qubit coupling efficiency, resonator-qubit coupling). For detailed comments, please see below.

The readout circuit is essentially the same as in Ref. [32]. We have added summary text to the supplement with appropriate references and an updated schematic, Fig. S5. For the referee's interest, see Section III.B. and Fig. S3 in the supplement of Ref. [32] (Arute et al., *Nature* 2019).

We have added the single-qubit and two-qubit gate durations (25 ns and 34 ns, respectively).

We have added definitions for the terms the reviewer pointed out.

Overall, this is a great result, and we shall recommend publication in Nature if the points raised above and the points detailed below are addressed carefully.

We thank the referee for their constructive comments, both these high-level criticisms and the detailed comments listed below. We appreciate their praise and recommendation for publication in *Nature* when their points are addressed.

Detailed comments on main text

Abstract: "... highlight the biggest challenges for future systems ...". Please specify what these are from your perspective. Otherwise, the statement is assertive while being very generic.

We feel this is phrased adequately as-is: this is an introductory paragraph, so it makes sense that we omit some conclusions and details. Even in the main text, we do not dwell on this topic for very long. We are leaving it alone for now and defer to the editor.

Abstract: "... the first experimental demonstration ...". My understanding is that claims of novelty are usually not allowed by editorial standards in most journals, even if correct.

We are happy to remove this if needed. We are leaving it alone for now and defer to the editor.

Section I, beginning of paragraph 1: The references are quoted in a very generic form. It would be useful to the reader to be more precise about what is covered in these references. Some of those present work that is very closely related to the current work, but the authors do not point this out. Others are covering different approaches to QEC. A bit more room for the discussion of the references would be helpful for the reader.

The referee is correct, although there are space constraints in how much we can explain and break out individual references. In this context, we're introducing related literature before digging into the specifics of our work, so it is not obvious how to highlight the distance-3 surface code experiments here without spending several sentences. We are leaving it alone for now and defer to the editor.

Section II, paragraph 2: Different terms are used for qubits which in the original literature were called ancilla qubits. We think it would be useful to help converge on a common terminology. The authors may want to argue for a suitable convention and help using it consistently. Terms currently used include ancilla, helper, measure, auxiliary qubits and possibly others.

We agree that there is some terminology variation in the literature that can be confusing. We are sticking with the "data qubit" and "measure qubit" parlance as in Ref. [36] (Fowler et al., *Phys. Rev. A* 2012) which we have also used in our other error correction experiments, such as Ref. [61] (Kelly et al., *Nature* 2015) and [42] (Chen et al., *Nature* 2021). Unfortunately, we do not believe there is room to editorialize on these naming conventions within the manuscript. We hope our consistent usage will help set the standard.

We specifically eschew the term "ancilla;" see for example the discussion by Wiesner (<https://arxiv.org/abs/1705.06768>). "Auxiliary" is another reasonable alternative, if a bit of a mouthful; we used it in a related context in Ref. [26] (Satzinger et al., *Science* 2021).

Section II, paragraph 3: It is pointed out that logical memory operation is most important. Equally important however, is the execution of gates between logical qubits. In that context, it would be adequate to also refer to D. Litinski, Quantum 3, 128 (2019) when citing Refs. [37, 38] for lattice surgery.

We agree with the referee that gates between logical qubits are essential. However, when these gates are conducted with lattice surgery, the sequence of events is essentially the same as running a memory experiment, just temporarily turning on stabilizers at the boundaries to “glue” qubits together and measuring lines of data qubits to “cut” them apart again. If you can do memory well, we believe you’re going to be able to do a lattice surgery CNOT well. This is the message we’re conveying in the sentence in question: *“Most surface code logical gates can be implemented by maintaining logical memory and executing different sequences of measurement.”* Eventually, with larger and higher-performance devices, we hope to see this validated experimentally.

We are also adding the reference to Litinski as suggested.

Figure 1c: The qubit idle errors marked DD seem to be significant but are discussed very little in the main text. Expanding a bit would help the reader appreciating the relevance of those errors.

The referee is correct to emphasize these idle errors. Substantial data qubit idle error during measurement (and in this case reset) is easy to overlook in a simulation, which can lead to misleading conclusions about thresholds. We are adding a clarifying remark to the Fig. 1c caption.

This comes up again towards the end of the paper in the discussion of Fig. 4a, where we budget about 20% of our $1/\Lambda_{3/5}$ to this idle error. We discuss it briefly there: *“This error budget shows that CZ error and data qubit decoherence during measurement and reset are dominant contributors.”*

Figure 1, caption b: “... focusing on one data qubit (gold) ...”. Stating more explicitly that the focus is on one data qubit (labeled with ψ) interacting with 4 measure qubits and one measure qubit (labeled with 0) interacting with four data qubits would render the statement more explicit.

We edited the caption as suggested.

Section II, last paragraph: The ZXXZ variant should be discussed in more detail, see general comments as well.

Please see above for ZXXZ discussion.

Also “This ensures that we do not preferentially measure even parities in the first few rounds of the code, which could artificially lower logical error rates due to bias in measurement error.” With respect to this statement, please refer explicitly to the experimental data in the supplementary figure S18 which demonstrates this bias and its effect on the logical error rates, otherwise this may be easily missed or the statement may appear poorly justified.

We are adding an explicit reference to the supplement as suggested. This point is a bit of a detail, but we wanted to include it in the main text as an “experimental best practice” to encourage adoption by the research community.

End of section III.: “We begin with a Pauli simulation based on the component error information in Fig. 1c in a probabilistic Clifford simulator, then incorporate coherence information, transitions to leaked states, and crosstalk errors ...”: We are surprised that the authors mention coherence information here. We assumed that the measured coherence times of the qubits are also used in the standard Pauli error model to model idle operations. Can the authors please clarify?

We have clarified the discussion regarding simulators. The mentioned text has been updated to: We begin with a depolarizing noise simulation based on the component error information in Fig. 1c, and then extend to a Pauli simulation with qubit-specific T_1 and $T_{2,\text{CPMG}}$, transitions to leaked states, and stray interactions between qubits during CZ gates (see supplement). We refer to this simulation as Pauli+.

The standard Pauli simulation did not incorporate coherence information directly into idle operations, since the average errors for idling were directly benchmarked and incorporated as depolarizing error.

Figure 2, caption a: “... diagonal pair ...”. In the literature and in the author’s own supplement they refer to these errors as space-time-like errors, which seems to be more commonly used in literature. Thus, we suggest to use this terminology here as well.

We are switching to “spacetime-like” as suggested.

Section IV, last paragraph: “We expect that mitigating leakage and stray interactions will become increasingly important as error rates decrease.” Can the authors please expand on what they mean by stray interactions. This would help readers to understand the main text.

The referee is correct that our explanation about “stray interactions” is thin, only referring to them as “stray” or “unwanted.” Unfortunately, we think due to length constraints there is not space to elaborate meaningfully in the main text. For clarity to the referee, we mean unwanted qubit state evolution due to unintended couplings, for example induced frequency shifts or entangling operations (like swapping or conditional phase interactions).

Section V, 2nd paragraph on page 5: “...leakage accumulation may cause distance-5 performance to degrade faster than distance-3 as logical error probability approaches 50%.”: Why is that? Is this due to the on average higher qubit-connectivity of the $d=5$ code? Do the authors have data for logical error probabilities approaching 50%?

This is a matter of speculation. We discuss it briefly earlier in the paper: “*We hypothesize that the greater increase in detection probability in the distance-5 code is due to increased stray interactions or leakage from simultaneously operating more gates and measurements.*”

We think this is indeed related to the higher qubit connectivity (the distance-5 experiment has a greater proportion of qubits with four active neighbors). Put simply, if leakage is accumulating

more rapidly in the distance-5 code, that should manifest in increasingly-many errors occurring, and its performance should degrade over the course of the long experiment.

We only have surface code data out to 25 rounds, as presented. At that point the logical error probability is already about 40%, getting close to a coin flip.

Section VI, paragraph 1: "... new dominant error mechanisms may arise ..." Please specify what type of errors you have in mind to be more precise.

Full sentence: *"Even as known error sources are suppressed in future devices, new dominant error mechanisms may arise as lower logical error rates are realised."*

Candidly, we cannot specify the error mechanisms, as they're specifically new and unknown, but one possibility could be novel stray interactions which are sub-dominant at the system scales we are currently exploring.

Also, "... 1D version of the surface code ..." We find referring to a repetition code in this way misleading. Please state that this is the repetition code. The repetition code is essentially attempting to correct errors in a conventional, classical approach.

Full context: *"To test the behaviour of codes with substantially lower error rates, we employ the bit-flip repetition code, a 1D version of the surface code. The bit-flip repetition code does not correct for phase flip errors and is thus unsuitable for quantum algorithms."*

We name it "repetition code" first and include a clarifying remark that it is unsuitable for quantum algorithms, so we do not feel there is anything misleading here. The parenthetical remark "a 1D version of the surface code" is merely meant to help draw a comparison, considering the repetition code as a $1 \times d$ surface code.

Section VI. A reference to Figure 4 b containing the relevant data is missing from section VI. Please add so that the panel will not be missed when reading the paper.

We added a reference.

Fig. 3c and corresponding text: The authors refer to chronologically improved performance. Is the data taken on the same device and what is the max. time span between data points?

Yes, all the experiments are on the same device and the same cooldown. The maximum time span between data points is about six weeks, and the minimum is three days.

Fig. 2a: Consider moving the red Z-error label a bit to the right to not overlap with the depiction of the surface code parity grid.

This is a good idea. Admittedly, it is tricky to get the perspective right in this figure. We have the red Z further to the left to clarify that there is no Hadamard on that qubit at the beginning of the cycle. The point is well taken, but we are leaving it unchanged for now.

Ref. [48] and [59] (McEwen et al., Removing ...) appear to be identical.

We fixed several duplication issues in the bibliography, including this one. We appreciate the attention to detail.

Detailed comments on Supplementary Information

XI. A.: "... with XY-4 phase cycling": Reference is missing.

We thank the referee for spotting this. We have added a reference to T. Gullion, D. Baker, M. S. Conradi, *J. Mag. Res.* **89**, 479-484 (1990).

XI. A.: "... the specific dynamical decoupling sequence is chosen on a per-qubit basis in order to ...": Can the authors elaborate on the adjustable parameters in their dynamical decoupling sequences?

We allow the number of pulses and hence the inter-pulse spacing to vary for each qubit addressed. We have added a sentence in this section to make this clear.

XI. C. and Fig. S7: The single-qubit gate duration is not stated.

We now state the gate durations in both the main text and in the supplement.

XI. D.: "CZ gates in the sycamore architecture arise from a state-selective dispersive shift ...": We find the term dispersive shift misleading here since the deployed CZ gates exploit a resonant interaction as far as we understood, not a dispersive one.

The referee understands correctly, and we have rewritten this section to more clearly state that the CZ is implemented via resonant swapping.

XI. D. and Fig. S8: The two-qubit gate duration is not stated. Furthermore, the coupling efficiency plotted in panel (b) is not defined. We assume it is defined with respect to the minimum achievable gate duration. Please clarify.

We now state the gate durations in both the main text and in the supplement. We have also added a definition of this coupling efficiency in the figure caption as suggested. To clarify further, it is the effective maximum qubit-qubit coupling efficiency during the CZ gate, based on our calibrations for the 34 ns duration. It is possible to implement faster or slower gates, thanks to the tunable coupling.

XI. E. and Fig. S9: Resonator-qubit coupling (panel b) is not defined. We assume this is the ratio between the dispersive coupling strength and the coupling strength if qubit and resonator were on resonance. Please clarify. Furthermore, it would be useful if the authors indicated not only the readout power in dBm in panel g but also the intra-resonator photon number or the conversion factor between the two.

We have added a definition for resonator-qubit coupling and switched out the readout power (in dBm) plot for one showing the maximum photon number. As alluded to by the referee, the photon number is a more meaningful number; however, our readout pulses are relatively short, meaning that the photon number changes drastically in time, which makes it difficult to give a

conversion factor. Instead, we have simulated the resonator responses given input power, resonator linewidth and frequency, and we report the maximum number of photons in each resonator during readout (which is not necessarily equal to the eventual state state number).

XI. G. 1.: "We can divide the corrections into ten groupings of compatible corrections for any distance of surface code, meaning that the corrections impact different detectors and can therefore be optimized simultaneously... Since the number of groupings is constant for any distance of surface code we expect this method to scale well as we increase the sizes of our system.": I find these statements hard to follow. Can the authors please explain in more detail?"

This section has now been expanded to clarify how the correction values are calibrated, and how this procedure can be done in parallel groupings.

XII. A. 3rd paragraph: "To avoid this issue, we decode an even subset of experimental trials by computing p_{ij} on the odd subset, and vice versa...": The authors mentioned earlier that they use a 25 cycle data set taken at the beginning for training the decoder. Was that data set thus not used to set the weights of the decoder? Please clarify.

We thank the referee for their attention to detail finding this inconsistency. The discussion in the decoding section is correct. Ultimately, the extra 25-cycle data set was not used to train the decoder except to test whether the decoder could be directly trained using previous data without significant degradation in performance, which is discussed later in the section. We have edited the earlier statement to avoid confusion.

XII. A. 4th paragraph: "We perform an abbreviated (due to the computational cost of the tensor network) distance-3 decoding experiment using 5, 9, and 13 rounds with this method." We appreciate that the authors performed such checks. In view of their comment we are curious about the computational cost of the tensor network decoding for $d=3$ and $d=5$. Can the authors give an idea about the computational cost (resources and time), potentially in the tensor network decoding section below?"

There is a huge gap between the efficiency of the tensor network decoder and the other decoders. The tensor network decoder was implemented on a cluster; it took tens of CPU years to decode the entire experiment. We have added this to the section.

XIII. 2nd paragraph: "... discarding the first round as it decreases the quality of our fits due to the unique nature of its errors, which disproportionately favor distance-5.": Do the authors understand why the $d=5$ experiment is favored more than the $d=3$ experiment when including the first round?"

To clarify this point, we have updated the main text:

"We start the fit at $t = 3$ to avoid two phenomena that advantage the larger code: the lower detection event fraction during the first cycle (see Fig. 2b, d), and the higher effective threshold caused by the confinement of errors to thin time slices [Dennis 2002]."

XIII. end of 2nd paragraph: "When combining data from multiple experiments, such as when considering data from multiple logical bases, this process is done on each dataset separately, and the final ϵ 's are averaged.": What about the five different basis states for each logical basis, are they fitted separately or jointly?"

We have clarified this in the text. To elaborate, the different initial states (which were randomly selected) are considered to be part of a single dataset.

XIII. last paragraph: "As a check that this can appropriately account for additional errors, we also fit an extended model in which the residuals in our fits come primarily from fluctuations in the logical error per cycle ϵ itself instead of binomial sampling noise." To allow us to judge this extended model in comparison to the binomial model, it would be helpful if the authors provided more details on the extended model.

We have now provided an explicit description of this extended model in the supplement.

XIV. A. 3. "Since our CZ gates are diabatic (implemented as two complete Rabi swaps)": We understood that the CZ gate used in this work is the one from Foxen et al. (2020), is this correct? Or why do the authors talk about two complete Rabi swaps rather than a single continuous round-trip in the 11-02 manifold?

Yes, the gate is as implemented in Foxen (2020). The phrase "two complete Rabi swaps" was meant as a simple explanation to contrast against an adiabatic gate. We appreciate the referee pointing out this source of confusion. The phrase now reads "implemented as one round trip swap $|11\rangle \rightarrow |02\rangle \rightarrow |11\rangle$."

XIV. A. 3. Last paragraph: "In order to accurately capture the above processes, we numerically calibrated then simulated the CZ gate matching the experimental qubit and coupler control schedules.": Was the numerical calibration of the gate in the simulation done using ideal waveforms or were potential waveform distortions included?

We have added the following clarification of the simulation. We included simple Gaussian filtering representative of our experimental setup. We did not include more deleterious distortions such as pulse settling, which we attempt to mitigate on the experiment side.

"Each transmon qubit was represented using the 5 lowest energy levels, and the coupler was represented using 4. Qubit idle frequencies (~6 GHz), hold frequencies, and gate durations were matched to experimental values, while the coupler frequency and qubit detuning at hold were the only optimized parameters [Foxen (2020)]. A Gaussian filter with 1 ns rise time was applied to all frequency waveforms."

We note that experimental data was taken on the device that confirmed both the predicted transition probabilities and conditional phases, but we do not include a discussion of this in the supplement. At the end of the paragraph, we have clarified how these simulation results are incorporated into the trajectories simulation:

"These effects are then combined into a single unitary operator U_{leakage} that is applied following the "ideal" CZ unitary. The ideal unitary acts as a perfect CZ in the computational subspace and as a local diagonal operator on the higher energy states, with phases determined by the gate time and qubit nonlinearities."

XIV. A. 4. "First, the inactive couplers (i.e. those not implementing a CZ gate) used to mediate the qubit-qubit couplings at idle are unable to cancel all effective couplings over all fixed excitation subspaces": Can the authors state the dominant residual couplings?

To clarify, we have added this to the end of the paragraph: “...the dominant error mechanisms stem from $|01\rangle \leftrightarrow |10\rangle$ transitions between neighboring qubits (undergoing separate CZ / idle gates). These transitions are highly sensitive to unwanted resonances arising during parallel operation.”

XIV. A. 5. “While the models above account for a majority of errors in the experiment, they typically under-predict the experimentally calibrated decay of fidelity.” I would add something like “... in benchmarking experiments” to avoid confusion with the decay of the logical fidelity. Furthermore, can the authors comment on the origin of the observed discrepancy? Is it due to control errors, additional dephasing upon tuning the qubits in frequency during two-qubit gates, etc.?

We have added the suggested clarification. It is difficult for us to speculate over the origin of the discrepancy with the benchmarking experiments (because there are quite a few possible sources), but we have added the following discussion to the end of this section:

“In all of these cases, effects such as microwave or DC (flux) control crosstalk may contribute to component fidelity decays. Effects involving other quantum degrees of freedom may also be involved, such as the coupler transitioning to its first excited state [Youngkyu (2021)] or coherent TLS that come close in frequency to a qubit. Drifts in control electronics or TLS frequencies between device calibration and collection of experimental data may also be involved. Measuring and mitigating these (and other) effects are an active area of research.”

XIV. B. 2.: To avoid confusion, it would be good if the authors stated at the beginning that they refer to the simulated qubit systems as devices throughout this section.

We have added a note to clarify this.

XIV. B. 2. 4th paragraph: “As a result, the full Hilbert space is separated into mutually incoherent leakage subspaces given by Kronecker products, ...” Why is it called leakage subspaces if the subspace can also be the computational subspace?

We have removed “leakage” to be more precise. The new sentence now reads:

“We assume that the states $|2\rangle$ and $|3\rangle$ are subject to rapid dephasing... The full Hilbert space is separated into mutually incoherent subspaces given by Kronecker products...”

XIV. B. 2. End of 4th paragraph: “As detailed below, approximating quantum error channels in terms of GP channels is premised on the assumption that the input is a highly entangled stabilizer state, ...” It is not clear to me why general Pauli channels should be premised on stabilizer states only. Can the authors please clarify?

We have removed the sentence as it is ambiguous and does not fit the context of the discussion in section XIV.B.2. The justification for using GP channels is discussed in more detail in section XIV.C.

XIV. B. 2. End of 5th paragraph: "Initial state of the qubits in U is traced over, while the final state for each qubit in D is a randomly selected $|0\rangle$ or $|1\rangle$ with equal probability." Doesn't one expect that the final state for qubits in D is more likely to be $|1\rangle$ than $|0\rangle$, simply because the leakage states first decays to $|1\rangle$?

The referee is correct; we are employing an approximation. Indeed, one does expect the D qubits to be mostly in state $|1\rangle$. A more accurate approximation would be to measure the expectation of Z Pauli for the qubits in D as a result of applying the noise channel. The transition to the computational subspace could then be modeled as probabilistically preparing the qubits in an eigenstate of Z (matching the Z expectation values). We have added some clarifying discussion in Section XIV.B.2, where the set D is defined:

"We note that a more physically accurate distribution could be used here. To account for the fact that transitions occur in the standard basis, we may instead randomly prepare the device in one of the σ_z eigenstates. In the context of the twirling approximation (Sec. XIV B 3), the probability of preparing one of the states could be set to match the device's $\langle\sigma_z\rangle$ observable, on the condition that the device decayed to the computational subspace."

XIV. B. 1. 3rd paragraph: "This means that no superposition is ever formed between the computational and leakage subspaces of any qubit, and we therefore are never required to keep more than two levels per qubit in our state vector." This seems like an important assumption and simplification. Can the authors comment on the consequence for the accuracy of the simulations?

We have added a justification in section XIV.B.1. We note that a complete discussion of this approximation is deferred to a later work (a paper is currently being written).

"This is justified by the fact that, of the three error mechanisms causing leakage transitions, two (heating and dephasing-induced leakage) are incoherent processes, while coherent leakage arising from crosstalk is rapidly "dephased" as the neighboring measure qubits are measured and reset. The system state is therefore well approximated as block diagonal with respect to the leakage and computational subspaces."

Fig. S19, caption: "... so that its standard deviation is bounded from above by $1/\sqrt{4N} \approx 0.11\%$ ": I didn't immediately realize that the factor four comes from assuming a binomial distribution for epsilon. Please consider adding this explanation.

We have revised the second sentence of the caption to explain: "Each data point is the logical error fraction over $N = 2 \times 10^5$ samples, which estimates the logical error probability p . This is modeled as a Bernoulli process with standard deviation $\sqrt{p(1-p)}$, so the standard deviation of the sample average is $\sqrt{p(1-p)/N} \leq 1/\sqrt{4N} \approx 0.11\%$ "

XV. A. "Finally, another imperfection in the device is the presence of unwanted interactions (crosstalk) between simultaneous gates (either single qubit idle or CZ), ...": What about crosstalk due to simultaneous single-qubit gates?

Next sentence: "This error channel is modelled individually for all elements and leads to CZ errors with a Pauli error probability (averaged over all CZs) of ..., which is roughly 15 % of the observed total CZ error": 1. The authors say that crosstalk for single qubit idling is also considered but only mention CZs gates here. Please clarify.

2. I understand that the crosstalk error stated comes from the simulations described in XIV. A. 4. Is this calculated value compatible with XEB fidelities for CZ gates measured individually and in parallel?

The primary origin of these unwanted interactions is the mode-mediated coupler design, making it heavily dependent on the frequencies of the qubit and coupler. During the CZ, the coupler frequency is much closer to the qubits, so stray couplings are much more sensitive compared with single-qubit gates or idling. This is why we primarily focus on unwanted interactions during the CZ. Based on the referee's feedback, we have modified the last paragraph in XV.1 to explain in more detail.

XV. B. and Fig. S22:

1. The authors state that the sensitivity analysis with respect to epsilon is distinct from the $1/\Lambda$ analysis around Fig. 4a. I find that the conclusion in this section is missing and that it would be helpful to better work out the relation to the error budget presented in Fig. 4a of the main text.

2. We notice that the components with largest absolute error rates have lowest sensitivity. Is this expected? Wouldn't it make more sense to look at the sensitivity with respect to relative changes in error rates, i.e. use $\frac{\Delta p_{\text{error}}}{p_{\text{error}}}$ instead of Δp_{error} ? At least an experimentalist rather thinks about gate error improvements on a relative scale.

The sensitivity study here is related but different from the one in Fig. 4a, as they were performed at different linearization points. There is no straightforward way to fully quantify the relationship between the two, as the exact functional forms of both logical error rate and $1/\Lambda$ remain unknown for realistic error models. Based on the referee's feedback, we have added Sec. XV.3 to discuss the relationship between two sensitivity studies and provide a conclusion.

It is interesting for the referee to notice that "the components with largest absolute error rates have lowest sensitivity." This is mostly true, with exception of leakage and data qubit idling. We do not have a simple explanation; it is not necessarily "expected." It does give us guidance on prioritizing future efforts. We have highlighted this in the concluding remarks in Sec. XV.3.

Fig. S24: The pixels in each of the cells have different widths and different heights, which makes it harder to interpret when zooming in.

We have improved this figure's resolution.

XVI. B. 2. 1st paragraph: "The number 0.043 written in the middle tile (5,5) is the average over rounds (specifically, the median) of the p_{ij} probabilities of all SX edges between these two X measure qubits.": Do the authors mean the average over cycles / time? If so, this could be said simpler.

Yes, the referee is correct that we mean averaging over time. We have rephrased this.

XVII 2nd to last paragraph: "we subtract the contribution of 9.5×10^{-4} from crosstalk and the contribution of $1.25 \times 2 \times 10^{-4}$ from CZ leakage.": It is not obvious to me where the factor of 1.25 in the leakage contribution comes from. Can the authors please clarify?

We thank the referee for raising the point: upon revisiting the analysis, we have concluded the factor of 1.25 was not needed. This factor was originally intended to account for the conversion between "average error" and "Pauli error." We have revised Fig. 4a and Table 3 accordingly,

including a few more small revisions. The CZ contribution increased modestly from 0.264 to 0.267.

Table IV: Ref. [24] is from year 2022.

We thank the referee for pointing this out. We had an inconsistency in the table between arXiv post year and publication year. We have updated this to be more consistent and specifically moved Zhao et al. after Krinner et al. in the table, both with 2022.

Response to Referee #3

In this work the authors have demonstrated suppression of quantum errors by scaling up the surface code from distance 3 to distance 5. Although the actual degree of suppression is small and breakeven has not yet been demonstrated, this work is a landmark in the field and, and I recommend it for publication. I particularly appreciate the balanced approach taken by the authors in writing the manuscript, wherein they have presented detailed error budgeting and identified necessary areas of improvement in the experiment.

We thank the referee for recommending our work for publication and their helpful comments and questions, which we have addressed below.

I do have some comments for the authors to consider below.

1. A lot of times some very technical points are mentioned in the main text without reference to the relevant section in supplementary material. Even adding a 1 sentence intuitive explanations for these concepts will make it much more accessible to Nature's audience.

We appreciate there are many technical concepts in this paper, and the referee is correct to emphasize making the main text as accessible as possible for *Nature's* audience. We include these insights and explanations where we can, although we are limited by length constraints. We address the referee's specific examples below.

One example is the statement "to avoid time-boundary effects that are advantageous to the distance-5 code". Only QEC experts (a smaller subset of QI community) will be familiar with the concept of time-boundary. A short 1-line explanation or reference will be useful here.

This is a good point. We have rephrased the sentence to explain further: *"We fit the logical fidelity $F = 1 - 2p_L$ to an exponential decay. We start the fit at $t = 3$ to avoid two phenomena that advantage the larger code: the lower detection event fraction during the first cycle (see Fig. 2b, d), and the higher effective threshold caused by the confinement of errors to thin time slices [Dennis 2002]."*

Another example is " This ensures that we do not preferentially measure even parities in the first few rounds of the code, which could artificially lower logical error rates due to bias in measurement error." More explanation of "bias in measurement error" is provided in the supplementary material although not explicitly cited at this point.

We have added an explicit reference to the supplement here.

2. The authors note that the leakage gets slightly worse when we move from distance 3 to distance 5 codes. They attribute this to the fact that more gates etc are being executed simultaneously in the dist-5 code. Can the authors say more about how they expect leakage to scale as the system size gets bigger than dist 5? Would it possible to mitigate stray interactions at the hardware level or is the next step to deal this with the error-correction architecture?

This is a good question, and it is difficult to forecast what might happen with larger devices.

One contribution is stray interactions between qubits, which seem to be fairly local in our devices. This advantages the smaller distance-3 code, where many of the qubits are near the boundary and have fewer than four active neighbors. As the code distance increases, a larger proportion of qubits are in the bulk. As a result, we expect this effect to saturate as we increase the code size. To the referee's direct question about leakage mitigation, this is an active area of research in device design and calibration, and we expect to make improvements, which we plan to discuss in future publications. Some small stray interactions will probably persist, but if they are sufficiently small and local, quantum error correction should handle them.

Another subject is classical control crosstalk, such as one qubit's control line influencing another qubit. The scaling here depends strongly on device architecture, such as the routing of control wiring and packaging. This is an ongoing research area to continue improving in hardware and testing at larger scales. It is also possible to mitigate with active cancellation, which has been demonstrated in many different works including this one, although that may become intractable at very large scales.

3. It would be useful to have a bit more details about the assumptions behind the simulations in Fig. 4d. As far as I understand, that figure ignores leakage. It would be nice to know how leakage will affect that plot. Or even how leakage must scale for that plot to hold.

The referee is correct that the simulations shown in Fig. 4c-d are Pauli simulations that do not include leakage. This is an excellent question and a subject of future research. The simulations were quite extensive ($\sim 10^9$ cycles per point to high distances over a 2D scan), so we chose a Pauli simulation for efficiency. There has been some published work on the effects of leakage at high distance (e.g., <https://arxiv.org/abs/1308.6642>). Ultimately, we expect to require some form of active leakage removal, either at the hardware level or built into the error-correction protocol, for the plot to hold its general shape. We expect that improving our leakage mitigation will become increasingly important as other error sources are reduced and we scale to even larger codes.

Reviewer Reports on the First Revision:

Referees' comments:

Referee #1 (Remarks to the Author):

The manuscript has much improved. The main text reads very well and the supplementary information is more complete, clear and informative. Overall, I find the responses of the authors satisfactory and their changes appropriate, except for a few points, see below.

I believe that the authors should consider all comments below and make appropriate (small) amendments before the paper can be published.

1. The thoughts on the issue with cosmic rays in your response are much appreciated. However, we note that the time-resolved work by the IBM team to which the authors refer, did not seem to report a change in T_1 , while this was the explicit focus of the previous study by the authors (drop in T_1 every 10 sec in <https://arxiv.org/abs/2104.05219>).

2. On finite size effects on the surface code. It is undoubtedly true that finite-size effects have been seen in simulation in many papers, this does not mean that their nature is well-understood (analytically) nor that results are consistent across simulations.

For example, in the color code study in Fig. 13 in <https://arxiv.org/pdf/2101.02211.pdf> one finds monotonic upwards behaviour of the circuit-level noise pseudo-threshold (error probability p such that $P_d(p)=p$ where P_d is the logical error probability for a distance d code) as well as monotonic upwards behaviour for the cross-over points (the probability where the logical error probabilities going from a d to $d+2$ code crosses). Such monotonic behaviour is what one expects for simple error models: larger d leads to a larger pseudo-threshold, larger codes are always better. In the previous papers to which the authors refer the data are not easy to parse/read visually (e.g. Fig. 10 in <https://arxiv.org/pdf/1202.5602.pdf>), and the behaviour does not seem consistent with monotonic-increasing behaviour (if the cross-over points are monotonically increasing, one cannot first be below this cross-over point so to decrease in logical error rate and then, at larger distances, be above it and increase in logical error rate).

I am not sure that adding [60] and the bound in Eq. (12) in [60] is at all helpful in understanding the dip in Fig. 4d (as Eq.(12) when plotted does not have such dip).

Possible explanations for deviations from monotonic-increasing behaviour could be due to the error model (e.g. some dominant error sources like CNOT errors lead to boundary checks of weight-2 being much more reliable than bulk checks and for small codes one can profit from such boundary effect?), or the process of decoding itself.

Even though this discussion goes beyond this paper, I believe that the authors should be cautious in claiming that their numerical behaviour is generic for the surface code as (1) it is not properly understood, (2) there is numerical data on similar codes (e.g. color codes) with different results.

Small things on main text:

1. Sentence "In addition to the expected pairs, we also quantify how often detection pairs occur which are unexpected in a local error model [19].": why reference [19], does this relate to FT studies

of concatenated codes which show that one can also deal with non-local errors?

2. Caption Fig. 4(b) two sentences "error decreases more rapidly with code distance. 50 cycles, .." seems a bit sloppy.

3. "high-distance logical error" in Section VI seems unclear to me ("high-weight error leading to logical error" seems accurate, but a logical error always has weight at least the distance, and distance is set by the code)

Suppl. Material

1. {\em idle} idle in Section XI B

2. Section XI B: cursive CZ versus non-cursive CZ

3. Section XI C: "qubit operating frequency" -> "qubit idle frequency"

4. Fig. S6: Would still be good to define measurement efficiency.

5. Section XI.H The new DEF is used in the text once and once in the legend of Fig.S12 but never defined explicitly. Reading the caption, it seems to be the detection probability averaged over measure qubits and cycles, so please define when you mention it.

In Fig. S12 the smoothed logical error probability seems to be able to go above 0.6, but $P_{\text{err}}(t)$ below Fig. S14 can not be more than 0.5 (random guess), so what is the issue here?

5. Section XIV B: Add space in Thekraus_sim library serves and bykraus_sim's quantum etc.

6. Section XIV B: "Details of this approximation are deferred to a future work." Not sure how useful this sentence is. Overall references to future work should be avoided (there are some other instances in the paper) unless it is quite certain that the team is working on this and it will appear (I cannot judge this).

7. Gottesmann-Knill -> Gottesman-Knill

8. Section XIV.B.2 on 2. measure describes the stabiliser update after a measurement in the Z basis of qubit j. This paragraph still has confusing text "Otherwise, use row transformafions ($\hat{G}_i \rightarrow \hat{G}_i \hat{G}_j$) to select a set of generators with either (a) \hat{G}_1 acting only on device j (i.e. it is unentangled), or (b) \hat{G}_1 and \hat{G}_2 the only generators acting on j (but also other devices), where $\hat{G}_1[j] = \sigma_x$ and $\hat{G}_2[j] = \sigma_z$." . Which set of generators is selected? The G_i , I presume, but then it refers to G_1 and G_2 , so only the G_1 and G_2 ? I know the idea of the procedure but I don't recognise it from the writing .

9. "heavily depends on the frequency configuration" -> "heavily depend on the frequency configuration"

10. "The contributions to $1/\lambda^{3/5}$ " -> "The contributions to $1/\lambda_{3/5}$ "

11. You write in Section XVI "The p_{ij} probability of an edge between nodes i and j is approximately equal to the sum of the probabilities of all error processes that produce clusters of detection events including the nodes i and j ".

Isn't the probability p_{ij} exactly equal to the probability for a basic error which causes defect i and j and nothing else (and this is why $-\log(p_{ij})$ is used as edge weight, similarly $p_{i,B}$ is the probability for a boundary edge error and when you have basic errors which cause a cluster of 3 errors you include a $p_{\{123\}}$ which you extract from the data in subsection C). Instead $\langle x_i x_j \rangle$ is the probability for defects at i and j and anything at the other nodes, so that sounds more like what you describe. The motivation of your sentence comes from the approximation that you describe in subsection C, but for conceptual clarity introducing the method by this approximation may not be the best.

12. Section XVI "Note that although use the ZXXZ variant of the surface code,..."

Referee #2 (Remarks to the Author):

We find that in general the authors have thoroughly addressed the reviewers' comments and that the quality of the manuscript has significantly improved. There remain a few unclear points which should be addressed before publication, though.

Main text:

Section V., 5th paragraph: "We start the fit at $t = 3$ to avoid two phenomena that advantage the larger code: the lower detection event fraction during the first cycle (see Fig. 2b, d), and the higher effective threshold caused by the confinement of errors to thin time slices [35]."

We are sorry to say that the added explanation does not clarify the points raised by Reviewers #2 and #3: First, Fig. 2b and 2d show close-to-identical detection event fractions for the $d=5$ and $d=3$ experiments for cycle number 0 and 1. Second, what does a higher effective threshold mean? There is only one error threshold for the surface code. Furthermore, the expression "confinement of errors to thin time slices" might be hard to understand even for experts. Please explain in more detail. The reference to a 50-page-long paper alone is not sufficient.

Supplementary Information and rebuttal letter:

Rebuttal letter response #2, page 4: "The referee is correct, although there are space constraints in how much we can explain and break out individual references. In this context, we're introducing related literature before digging into the specifics of our work, so it is not obvious how to highlight the distance-3 surface code experiments here without spending several sentences. We are leaving it alone for now and defer to the editor."

We emphasize the importance of setting previous work into context with the new work. Breaking

out the references into smaller sets at this location allows the reader to better judge the achievements of the work and helps the reader to look up specific references. A simple break out with a minor increase in word count could be Several works have reported quantum error correction on codes able to correct a single error, including the distance-3 color [20], surface [22, 24], and heavy-hexagon subsystem [23] codes. Single-cycle experiments focused on logical state preparation and measurement [21,25,26]. Continuous variable codes offer an alternative approach to quantum error correction [27-30].

Rebuttal letter response #2, page 9; Fig. S6 and caption: “and we report the maximum number of photons in each resonator during readout (which is not necessarily equal to the eventual state state number).”

Typo in the last three words, which makes it hard to understand the sentence. Do the authors mean steady-state photon number or refer to the qubit-state dependence of the photon number? Please clarify.

Fig. S12: x-axis label is incomplete.

Fig. S4a: We noticed that the interaction frequency for the two-qubit gates is not defined. There is no obvious definition for a CZ gate, so please clarify.

Section XIII., last paragraph: “We fit both the mean and the variance, where the weights for the least-squares fit in this model come from these fluctuations in ϵ , ...”

We appreciate that the authors added more information on the extended model. It is not very explicit though. Are we correct to assume that the authors first fit the mean ϵ to the decay of the logical fidelity and in a second step the variance σ_{ϵ} to the variance of the fidelity? Also, please be explicit how you determine the weights of (both?) fits. Furthermore, we wonder why the authors did not determine error bars from a bootstrapping method.

Rebuttal letter, response #1, page 21; Section XIV 2nd paragraph; Section XIV B. 1.: Reviewer #1 correctly points out that quantum trajectory simulations have been done in other papers in the surface code context. The authors added related references, but a reference to [22] Krinner et al., in which quantum trajectory simulations are presented, is missing.

Section XIV. A. 4.: “Details of these simulations are deferred to a later work, though we note that the dominant error mechanisms stem from $|01\rangle \leftrightarrow |10\rangle$ transitions between neighboring qubits (undergoing separate CZ / idle gates). These transitions are highly sensitive to unwanted resonances arising during parallel operation.”

In the last sentence, it is not clear what type of resonances the authors refer to. In fact, we cannot make sense of a transition (being itself a resonance) being sensitive to a resonance in that context.

Author Rebuttals to First Revision:

Referee #1

*The manuscript has much improved. The main text reads very well and the supplementary information is more complete, clear and informative. Overall, I find the responses of the authors satisfactory and their changes appropriate, except for a few points, see below.
I believe that the authors should consider all comments below and make appropriate (small) amendments before the paper can be published.*

We thank the referee for their positive assessment and additional feedback, which we have carefully considered and used to improve the manuscript.

1. The thoughts on the issue with cosmic rays in your response are much appreciated. However, we note that the time-resolved work by the IBM team to which the authors refer, did not seem to report a change in T_1 , while this was the explicit focus of the previous study by the authors (drop in T_1 every 10 sec in <https://arxiv.org/abs/2104.05219>).

We agree with the referee and look forward to learning more about such results in the future.

2. On finite size effects on the surface code. It is undoubtedly true that finite-size effects have been seen in simulation in many papers, this does not mean that their nature is well-understood (analytically) nor that results are consistent across simulations.

For example, in the color code study in Fig. 13 in <https://arxiv.org/pdf/2101.02211.pdf> one one finds monotonic upwards behaviour of the circuit-level noise pseudo-threshold (error probability p such that $P_d(p)=p$ where P_d is the logical error probability for a distance d code) as well as monotonic upwards behaviour for the cross-over points (the probability where the logical error probabilities going from a d to $d+2$ code crosses). Such monotonic behaviour is what one expects for simple error models: larger d leads to a larger pseudo-threshold, larger codes are always better. In the previous papers to which the authors refer the data are not easy to parse/read visually (e.g. Fig. 10 in <https://arxiv.org/pdf/1202.5602.pdf>), and the behaviour does not seem consistent with monotonic-increasing behaviour (if the cross-over points are monotonically increasing, one cannot first be below this cross-over point so to decrease in logical error rate and then, at larger distances, be above it and increase in logical error rate).

I am not sure that adding [60] and the bound in Eq. (12) in [60] is at all helpful in understanding the dip in Fig. 4d (as Eq.(12) when plotted does not have such dip). Possible explanations for deviations from monotonic-increasing behaviour could be due to the error model (e.g. some dominant error sources like CNOT errors lead to boundary checks of weight-2 being much more reliable than bulk checks and for small codes one can profit from such boundary effect?), or the process of decoding itself. Even though this discussion goes beyond this paper, I believe that the authors should be cautious in claiming that their numerical behaviour is generic for the surface code as (1) it is not properly understood, (2) there is numerical data on similar codes (e.g. color codes) with different results.

We appreciate the referee's concern that finite-size effects are not well-understood and indeed agree that results are frequently not consistent across simulations. As the referee points out, this can be due to the decoder, the error model, boundary stabilizer effects, the code chosen, and so on. One major choice is in the definition of the logical error rate. For example, we could replot our data by redefining the logical error rate to be per d -cycles rather than per cycle by

transforming each data point according to $p_{\text{new}} = (1 - (1 - 2 p_{\text{old}})^d) / 2$. The per-d-cycles metric, for example, is used in Figure 11 of Beverland et al. (<https://arxiv.org/abs/2101.02211>), which similarly shows crossover points ascending monotonically upward.

Under this transformation, we can compare the per-cycle logical error rate graph on the left to the per-d-cycles logical error rate graph on the right, where the x-axis again labels the scalar multiple of our XEB error rates as in Figure 4.

On the left-hand side, one observes that the crossings between code distances approach the threshold, which appears higher but drifts left, from above. This defines the crossover regime we refer to in the paper. On the right-hand side, the per-d-cycles redefinition causes the crossovers to approach the threshold, which appears lower but drifts right, from below. This is consistent with the linked paper, and has been discussed before in the nice paper on finite-size effects on thresholds by Stephens (<https://arxiv.org/abs/1311.5003>). Figure 13 seems to use a different ansatz for defining these crossovers, but all this is to say that definitional inconsistencies can also influence the finite-size effect behavior.

In summary, the referee is correct that several factors (including just the definition of the logical error rate) play into an individual simulation's behavior. However, we believe:

- 1) The simulation is correct: this is a behavior that can occur, and has been observed independently in other simulations.
- 2) Because this behavior can occur, we prefer not to make the stronger claim of being below threshold (unless the distance-5 code were much better than distance-3). We are more comfortable with calling this the crossover regime, as it is a simple empirical claim.

We have updated the text to reflect this discussion, including a reference to Stephens. We have also removed the reference to Fowler [60], which the referee found unhelpful.

Small things on main text:

1. Sentence "In addition to the expected pairs, we also quantify how often detection pairs occur which are unexpected in a local error model [19].": why reference [19], does this relate to FT studies of concatenated codes which show that one can also deal with non-local errors?

Here, we were looking for the first reference for the type of error model which is generated "locally," where error channels are supported on all the qubits to which a gate is applied. We have removed the reference and clarified the sentence.

2. Caption Fig. 4(b) two sentences "error decreases more rapidly with code distance. 50 cycles, ..." seems a bit sloppy.

We have converted the sentence fragment into a full sentence.

3. "high-distance logical error" in Section VI seems unclear to me ("high-weight error leading to logical error" seems accurate, but a logical error always has weight at least the distance, and distance is set by the code)

We have rewritten the sentence to clarify that we are referring to logical errors in the higher-distance codes, for example $d > 15$.

Suppl. Material

We thank the reviewer for these detailed comments; we have fixed them all. Please see below for some additional discussion on a few points.

1. {em idle} idle in Section XI B
2. Section XI B: cursive CZ versus non-cursive CZ
3. Section XI C: "qubit operating frequency" -> "qubit idle frequency"
4. Fig. S6: Would still be good to define measurement efficiency.

We have added more explanation and two references.

5. Section XI.H The new DEF is used in the text once and once in the legend of Fig.S12 but never defined explicitly. Reading the caption, it seems to be the detection probability averaged over measure qubits and cycles, so please define when you mention it.

This was a mistake. "DEF" means "detection event fraction," but we have replaced this with "detection probability p_d " to be consistent, both in the figure and in the text.

In Fig. S12 the smoothed logical error probability seems to be able to go above 0.6, but $P_{err}(t)$ below Fig. S14 can not be more than 0.5 (random guess), so what is the issue here?

The referee is correct that generally we would expect the "worst case" logical performance to be a 50/50 random guess, while the moving averages in Fig. S12 jump as high as 0.6 error probability. We do not believe this is cause for concern.

First, consider the statistical fluctuations: each plotted point is an average of just 30 bits, so for a 50/50 distribution we would expect a standard deviation of about 0.1 around the mean of 0.5. Additionally, there are only about 100 samples in the region of interest, corresponding to a standard deviation of about 0.05. The excess above 0.5 may simply be statistical in nature.

Furthermore, it's conceivable that the logical measurement could become so biased by the high-energy event that the logical error probability could indeed exceed 0.5. Consider the case where we prepare $|1\rangle^{\otimes 25}$ on the data qubits. If the high-energy event biases the measurements to result in $|0\rangle$ (for example from very low T_1), we could see logical errors most of the time.

5. Section XIV B: Add space in Thekraus_sim library serves and bykraus_sim's quantum etc.

6. Section XIV B: "Details of this approximation are deferred to a future work." Not sure how useful this sentence is. Overall references to future work should be avoided (there are some other instances in the paper) unless it is quite certain that the team is working on this and it will appear (I cannot judge this).

Members of our team are actively working towards a separate paper on these simulation details. However, we agree with the referee that these remarks do not add much, so we have removed them.

7. Gottesmann-Knill -> Gottesman-Knill

8. Section XIV.B.2 on 2. measure describes the stabiliser update after a measurement in the Z basis of qubit j. This paragraph still has confusing text "Otherwise, use row transformations ($\hat{G}_i \rightarrow \hat{G}_i \hat{G}_j$) to select a set of generators with either (a) \hat{G}_1 acting only on device j (i.e. it is unentangled), or (b) \hat{G}_1 and \hat{G}_2 the only generators acting on j (but also other devices), where $\hat{G}_1[jj] = \sigma_x$ and $\hat{G}_2[jj] = \sigma_z$." Which set of generators is selected? The G_i , I presume, but then it refers to G_1 and G_2 , so only the G_1 and G_2 ? I know the idea of the procedure but I don't recognise it from the writing .

We mean that we're selecting a set of generators so that, without loss of generality, G_1 (and potentially G_2) are used to measure the qubit according to the above prescription. We have added "without loss of generality" to the text to clarify that G_1 and G_2 are not distinguished.

9. "heavily depends on the frequency configuration" -> "heavily depend on the frequency configuration"

10. "The contributions to $1/\Lambda^{3/5}$ " -> "The contributions to $1/\Lambda_{\{3/5\}}$ "

11. You write in Section XVI "The p_{ij} probability of an edge between nodes i and j is approximately equal to the sum of the probabilities of all error processes that produce clusters of detection events including the nodes i and j ".

Isn't the probability p_{ij} exactly equal to the probability for a basic error which causes defect i and j and nothing else (and this is why $-\log(p_{ij})$ is used as edge weight, similarly $p_{i,B}$ is the probability for a boundary edge error and when you have basic errors which cause a cluster of 3 errors you include a $p_{\{123\}}$ which you extract from the data in subsection C). Instead $\langle x_i x_j \rangle$ is the probability for defects at i and j and anything at the other nodes, so that sounds more like what you describe. The motivation of your sentence comes from the approximation that you describe in subsection C, but for conceptual clarity introducing the method by this approximation may not be the best.

This is an excellent question and a subtle point. In this section, we are concerned with surface code diagnostics. To make these diagnostics digestible, we restrict to at most two-body correlations. For the purposes of decoding, these values would be "over-weighted" by error events that trigger higher-weight configurations of nodes that include i and j , among others.

The referee is absolutely right that when configuring the decoder, we would prefer to weight an edge between nodes i and j using the probabilities for errors that trigger those two nodes exactly. In the case of generalized p_{ij} , including higher body correlations helps bring us closer to this ideal situation, where we "remove" extra error probability caused by certain higher-order correlations. We have added a new sentence to help clarify.

12. Section XVI "Note that although use the ZXXZ variant of the surface code,.."

Referee #2

We find that in general the authors have thoroughly addressed the reviewers' comments and that the quality of the manuscript has significantly improved. There remain a few unclear points which should be addressed before publication, though.

We thank the referee for their positive assessment and additional feedback, which we have carefully considered and used to improve the manuscript.

Main text:

Section V., 5th paragraph: "We start the fit at $t = 3$ to avoid two phenomena that advantage the larger code: the lower detection event fraction during the first cycle (see Fig. 2b, d), and the higher effective threshold caused by the confinement of errors to thin time slices [35]."

We are sorry to say that the added explanation does not clarify the points raised by Reviewers #2 and #3: First, Fig. 2b and 2d show close-to-identical detection event fractions for the $d=5$ and $d=3$ experiments for cycle number 0 and 1. Second, what does a higher effective threshold mean? There is only one error threshold for the surface code. Furthermore, the expression "confinement of errors to thin time slices" might be hard to understand even for experts. Please explain in more detail. The reference to a 50-page-long paper alone is not sufficient.

We apologize for not being clearer. We have expanded the explanation in the text. To elaborate further, we believe there are two effects at play here.

First, the reviewer is correct that there are close-to-identical detection probabilities for the $d=5$ and $d=3$ experiments during cycle 0. However, this close-to-identical value is significantly lower than in future cycles.

For cycles 1 through $n - 1$, detectors are formed by comparing two measurements obtained during two successive cycles of syndrome extraction. However, in cycle 0, detectors are formed from only one measurement obtained from one cycle of syndrome extraction, along with our knowledge of the initialized qubit pattern.

One can interpret this as saying that, during the initial cycle, the error rate is "turned down" relative to later cycles. At this lower effective error rate, the gap between the distance-5 and distance-3 logical error probabilities will widen due to the increased-distance protection.

Second, the referee is correct to point out that there is of course only one true threshold for the surface code. This is obtained by achieving error suppression in the limit of distance $\rightarrow \infty$ while allowing the number of cycles $\rightarrow \infty$. However, we could instead consider fixing the number of cycles to some small constant and only taking the limit of distance $\rightarrow \infty$. We expect the threshold of this hypothetical experiment to be higher because it is more difficult for error strings to find their way across the lattice, being confined to a smaller number of time slices.

In the extreme case of having only a single time-slice of detectors, we obtain something approaching the so-called "code-capacity" (i.e. 2D) model of surface code error correction, with a threshold more than 3 times that of the "phenomenological" (i.e. 3D) model of surface code

error correction. One could imagine being well above the “true” threshold of the surface code, but below this code-capacity threshold. In that case, the logical error probability of the lower distance code will be lower after many cycles, but the logical error probability of the higher distance code could be initially lower (after only a few cycles). In fact, this is indeed what we observed in earlier, noisier experiments.

In summary, we believe that both the lower “effective” error rate and the higher “effective” threshold obtained during the single-cycle experiment noticeably advantage the distance-5 code over the distance-3 code, and so we exclude it from our fit. We have edited the text to try to clarify these points.

Supplementary Information and rebuttal letter:

Rebuttal letter response #2, page 4: “The referee is correct, although there are space constraints in how much we can explain and break out individual references. In this context, we’re introducing related literature before digging into the specifics of our work, so it is not obvious how to highlight the distance-3 surface code experiments here without spending several sentences. We are leaving it alone for now and defer to the editor.”

We emphasize the importance of setting previous work into context with the new work. Breaking out the references into smaller sets at this location allows the reader to better judge the achievements of the work and helps the reader to look up specific references. A simple break out with a minor increase in word count could be Several works have reported quantum error correction on codes able to correct a single error, including the distance-3 color [20], surface [22, 24], and heavy-hexagon subsystem [23] codes. Single-cycle experiments focused on logical state preparation and measurement [21,25,26]. Continuous variable codes offer an alternative approach to quantum error correction [27-30].

We thank the reviewer for their concrete suggestion and the editor for the latitude with space. We have adopted this expanded explanation of the references with some refinements.

Rebuttal letter response #2, page 9; Fig. S6 and caption: “and we report the maximum number of photons in each resonator during readout (which is not necessarily equal to the eventual state state number).”

Typo in the last three words, which makes it hard to understand the sentence. Do the authors mean steady-state photon number or refer to the qubit-state dependence of the photon number? Please clarify.

This was a mistake; we meant “steady-state.” Some of our measurement pulses are relatively short in duration, so the actual photon occupation does not reach a steady-state value.

Fig. S12: x-axis label is incomplete.

We have fixed this.

Fig. S4a: We noticed that the interaction frequency for the two-qubit gates is not defined. There is no obvious definition for a CZ gate, so please clarify.

We have added a definition, the mean of qubits’ frequency (f_{10}) values in the middle of the CZ.

Section XIII., last paragraph: “We fit both the mean and the variance, where the weights for the least-squares fit in this model come from these fluctuations in ϵ , ...”
We appreciate that the authors added more information on the extended model. It is not very explicit though. Are we correct to assume that the authors first fit the mean ϵ to the decay of the logical fidelity and in a second step the variance σ_{ϵ} to the variance of the fidelity? Also, please be explicit how you determine the weights of (both?) fits. Furthermore, we wonder why the authors did not determine error bars from a bootstrapping method.

Bootstrapping is a technique to estimate variance in the estimate arising from statistical fluctuations, so would not capture the additional variance we needed to capture due to device drift. Propagation of binomial errors is a linear approximation to bootstrapping, which will be a very good approximation with the number of statistics we collected.

The weights for fits in the fluctuating-logical-error model are given by the inverse of the variance that the model assigns to the data points (by inverting the data point at a particular cycle to obtain the mean logical error rate). This variance depends on the value we estimate for σ_{ϵ} , but it varies uniformly across all the data points, so changing the estimate for σ_{ϵ} does not affect the fit for the mean epsilon. Having fit the mean epsilon, we then fit σ_{ϵ} .

Rebuttal letter, response #1, page 21; Section XIV 2nd paragraph; Section XIV B. 1.: Reviewer #1 correctly points out that quantum trajectory simulations have been done in other papers in the surface code context. The authors added related references, but a reference to [22] Krinner et al., in which quantum trajectory simulations are presented, is missing.

We have added this reference.

Section XIV. A. 4.: “Details of these simulations are deferred to a later work, though we note that the dominant error mechanisms stem from $|01\rangle \leftrightarrow |10\rangle$ transitions between neighboring qubits (undergoing separate CZ / idle gates). These transitions are highly sensitive to unwanted resonances arising during parallel operation.”
In the last sentence, it is not clear what type of resonances the authors refer to. In fact, we cannot make sense of a transition (being itself a resonance) being sensitive to a resonance in that context.

We have updated the wording to clarify this sentence. By “transitions,” we mean swapping between $|01\rangle$ and $|10\rangle$. This can be affected directly by those energy levels colliding as well as effects from other higher states, which are the “resonances” we refer to.